# Robust and Personalized Federated Learning with Spurious Features: an Adversarial Approach

## Abstract

A common approach for personalized federated learning is fine-tuning the global machine learning model to each local client. While this addresses some issues of statistical heterogeneity, we find that such personalization methods are often vulnerable to *spurious features*, leading to bias and diminished generalization performance. However, debiasing the personalized models under spurious features is difficult. To this end, we propose a strategy to mitigate the effect of spurious features based on our observation that the global model in the federated learning step has a low accuracy disparity due to statistical heterogeneity. Then, we estimate and mitigate the *accuracy disparity* of personalized models using the global model and *adversarial transferability* in the personalization step. We theoretically establish the connection between the adversarial transferability and the accuracy disparity between the global and personalized models. Empirical results on MNIST, CelebA, and Coil20 datasets show that our method reduces the accuracy disparity of the personalized model on the bias-conflicting data samples from 15.12% to 2.15%, compared to existing personalization approaches, while preserving the benefit of enhanced average accuracy from fine-tuning.

## 1 Introduction

Federated learning (FL) is a leading framework for clients to collaboratively train a shared global machine learning (ML) model without releasing their local private datasets (McMahan et al., 2017; Kairouz et al., 2019). The jointly trained global model could be further fine-tuned on each client's local dataset to produce personalized models (Fallah et al., 2020; T. Dinh et al., 2020; Li et al., 2021). While existing theoretical and empirical results highlight how personalized models improve accuracy on local data, few works consider what features the personalized models learn from the local dataset. Our motivating hypothesis is that *not all local features are beneficial*.

For example, consider a gender prediction task using face images, where the ML model learns to predict gender based on hair color because females are more likely to have blond hair (Sagawa et al., 2020). In this case, the hair color is called a *spurious feature* because it only statistically correlates with the gender on biased-aligned samples but not necessarily on the overall population. Thus, the accuracy of a model that relies on spurious features such as hair color is likely to drop significantly on bias-conflicting samples where the spurious correlation does not hold, e.g., for blond male images (Sagawa et al., 2020). This paper calls the accuracy difference of an ML model on the dataset with spurious features and the dataset without spurious features *accuracy disparity*. More broadly, the accuracy disparity caused by spurious features leads to issues in both fairness (McNamara et al., 2019; Zhao & Gordon, 2019; Agarwal et al., 2019; Chi et al., 2021), i.e., racial bias (Khani & Liang, 2021) and robustness, i.e., accuracy decrease under distribution shift (Zhao et al., 2019; Koh et al., 2021). Compared to the global model, because of the local fine-tuning, the personalized models are more vulnerable to spurious features and have a larger accuracy disparity.

Empirically, we observe that the typical non-i.i.d. data distributions in FL settings reduce the accuracy disparity of the global model. One potential explanation is that the statistical heterogeneity (Wang

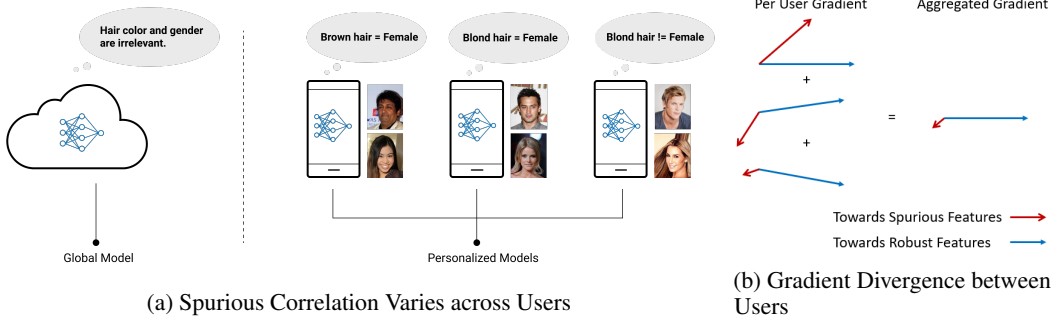

(a) Spurious Correlation Varies across Users

(b) Gradient Divergence between Users

Figure 1: The statistical heterogeneity of spurious features leads to diverse gradients. The statistical heterogeneity leads to a global ML model with a low accuracy disparity in a federated learning setting because the aggregation step in the central server averages out the gradients resulted from local spurious features.

et al., 2021) under spurious features across users is larger than that of non-spurious features. As a result, the aggregation operation in the central server averages out the diverse shifts under the spurious features (Figure 1) so that the global model becomes more robust against spurious features on the benchmark datasets. On the other hand, the local fine-tuning step will result in a personalized model that entangles spurious features and becomes biased. Hence, it remains a key challenge how to debias the personalized models.

Various methods (Wang et al., 2019; Sagawa et al., 2020; Liu et al., 2021) have been developed to disentangle spurious features from ML models, but few can be easily applied to personalized models under the federated learning setting. The main reason lies in their reliance on bias-conflicting samples. The bias-conflicting samples can be rare. For example, in the CelebA dataset, only around 1.7k samples are bias-conflicting out of more than170k samples. If we distribute the CelebA dataset across users, nearly half do not have any bias-conflicting samples. Prior work (Li & Wang, 2019) has tried using a global proxy dataset for training FL models. However, a global proxy dataset may not characterize the local samples well. Furthermore, estimating the accuracy disparity of the personalized model becomes difficult without access to bias-conflicting samples.

To approach this problem, we propose a novel method to reduce the accuracy disparity of personalized models. Our proposed method does not rely on the bias-conflicting samples, e.g., blond male images in the aforementioned task of gender prediction, which are often rare and may not be available for every user. Instead, inspired by prior works (Tramèr et al., 2017; Liang et al., 2021) on the transferability of adversarial examples, we use the global model as a reference and the *adversarial transferability* between the global model and the personalized models as a proxy to estimate the accuracy disparity of the personalized models. The intuition is that if two ML models use disjoint subsets of features, the adversarial examples that one ML model generates do not transfer to the other ML model. Based on this intuition, we propose the following hypothesis:

> *If the personalized models entangle spurious features (thus increasing the accuracy disparity), the adversarial examples generated by the global model (that uses non-spurious features) do not transfer to the personalized models.*

Empirically, we show that the adversarial transferability between the global and personalized models strongly correlates with the accuracy disparity between the global and personalized models (Section 4), validating our hypothesis. We further theoretically connect the adversarial transferability to the accuracy disparity (Section 5). Based on the empirical observations and theoretical results, we develop a method that enforces adversarial transferability between the global and the personalized models to reduce the accuracy disparity of personalized models (Section 6). Our contributions are summarized as follows:

- We empirically evaluate the accuracy disparity of the global and personalized models in a federated learning setting with spurious features, highlighting a risk of existing personalization methods.
- We design a method to estimate the accuracy disparity of the personalized models, based on the low accuracy disparity global model and the adversarial transferability between the global and personalized models.

- We theoretically connect the adversarial transferability and the accuracy disparity of the global and personalized models.
- We develop a methods to reduce the accuracy disparity of personalized models by enforcing the adversarial transferability between the global and personalized models.

Empirically, we conduct extensive experiments to validate the effectiveness of the proposed methods in mitigating accuracy disparity under the FL setting. Our experiments on MNIST (Deng, 2012), CelebA (Liu et al., 2015), and Coil20 (Nene et al., 1996) datasets show that the proposed approach reduces the accuracy disparity of personalized models from $15.12\%$ to $2.15\%$, which is closer to that of the global model ($-0.63\%$). Our method also preserves the benefit of the enhanced average accuracy from fine-tuning, resulting in $3.43\%$ accuracy improvement on the biased test set and $0.85\%$ accuracy improvement on the biased-conflicting test set.

## 2 RELATED WORK

**Personalized Federated Learning**   Fine-tuning is typical for personalization methods. The meta-learning-based method first trains a global model and fine-tunes the global model locally (Fallah et al., 2020). Other methods using multi-task learning (Li et al., 2021) or Moreau envelopes (T. Dinh et al., 2020) have an interpretation as fine-tuning the local model along with training the global model. Fine-tuning is also compatible with clustering-based method (Ghosh et al., 2020).

**Debiasing ML Models**   A few prior works (Li & Vasconcelos, 2019; Sagawa et al., 2020) utilize group labels, which might require human annotation, to debias ML models. For example, the group distributional robust optimization (DRO) method (Sagawa et al., 2020) aims to optimize the worst-case error rate of ML models across different (often manually annotated) groups. Some groups contain bias-conflicting samples while others do not. Residual learning-based methods (He et al., 2019; Nam et al., 2020; Liu et al., 2021) train a biased ML model and up-weight the residual, which mainly contains bias-conflicting samples that the biased ML model mis-predicts. Chi et al. (2021) aims to mitigate the accuracy disparity in regression problems via learning the appropriate representations. However, all these methods rely on the explicit access to the bias-conflicting samples, making them difficult to apply on personalized federated learning, where bias-conflicting samples may not be accessible for every client.

## 3 PRELIMINARIES

**Definitions and Notation**   A data sample is a vector $\boldsymbol{x} = [\boldsymbol{x}_r, \boldsymbol{x}_s]$, where $\boldsymbol{x}_r$ corresponds to the robust and non-spurious features and $\boldsymbol{x}_s$ are the spurious features. $d_s$ is the dimension of spurious features. Let $y$ be a label, and define $\ell : \mathcal{Y} \times \mathcal{Y} \to \mathbb{R}$ to be a $\lambda$-smooth, twice differentiable loss function and $\mathcal{L}(f, \mathcal{D}) = \mathbb{E}_{(\boldsymbol{x},y) \sim \mathcal{D}}[\ell(f(\boldsymbol{x}), y)]$ to be the empirical risk. $\boldsymbol{w}_g$ and $\boldsymbol{w}_p$ are the weights for the global model $f_g : \mathcal{X} \to \mathcal{Y}$ and personalized model $f_p : \mathcal{X} \to \mathcal{Y}$, respectively. $\gamma$ is the ratio between the gradient norms of the global and personalized models, $\frac{\|\nabla_{\boldsymbol{x}}\ell(f_g(\boldsymbol{x}),y)\|}{\|\nabla_{\boldsymbol{x}}\ell(f_p(\boldsymbol{x}),y)\|}$. Let $\mathrm{supp}(\mathcal{D})$ be the support of distribution $\mathcal{D}$. We define a global data distribution $\mathcal{D}_g$, a biased local data distribution $\mathcal{D}_b$, and assume a bias-conflicting local data distribution $\mathcal{D}_{bc}$. We define the "pseudo-gradient" as the difference between the updated local model and the global model from the previous round (sometimes we will use the term "gradient" when it is clear from context). $\langle \cdot, \cdot \rangle$ denotes an inner product of two vectors and $\cdot \frown \cdot$ denotes a concatenation of two vectors. $\theta$ is the angle between $\nabla_{\boldsymbol{x}}\ell(f_g(\boldsymbol{x}), y)$ and $\nabla_{\boldsymbol{x}}\ell(f_p(\boldsymbol{x}), y)$. $\theta_g$ is the angle between $\nabla_{\boldsymbol{x}}\ell(f_g(\boldsymbol{x}), y)$ and $\nabla_{\boldsymbol{x}_r}\ell(f_g(\boldsymbol{x}), y) \frown \boldsymbol{0}$, which measures the entanglement of the global model to spurious features.

**Training and Personalization Methods**   We train the global model using the federated averaging algorithm (McMahan et al., 2017), which learns a model $f : \mathcal{X} \to \mathcal{Y}$ that minimizes: $\mathcal{L}(f, \mathcal{D}_g) = \sum_{i=1}^{N} |\mathcal{D}_{b_i}|/|\mathcal{D}_g| \cdot \mathbb{E}_{\boldsymbol{x},y \sim \mathcal{D}_{b_i}} \ell(f(\boldsymbol{x}), y)$, where $N$ is the number of clients, $\mathcal{D}_{b_i}$ is the biased local dataset for client $i$, and $\mathcal{D}_g = \cup_{i=1}^{N} \mathcal{D}_{b_i}$ is the global dataset. When the global model $f_g$ converges, $f_g$ will be sent to local clients for further fine-tuning by minimizing $\mathbb{E}_{\boldsymbol{x},y \sim \mathcal{D}_{b_i}} \ell(f(\boldsymbol{x}), y)$.

**Adversarial Examples and Transferability**   We can generate an adversarial example $\boldsymbol{x}_{adv}$ given data sample $\boldsymbol{x}$ with label $y$ by solving:

$$\boldsymbol{x}_{adv} = \underset{\|\boldsymbol{x}' - \boldsymbol{x}\| \leq \epsilon}{\arg\max} \, \ell(f(\boldsymbol{x}'), y), \tag{1}$$

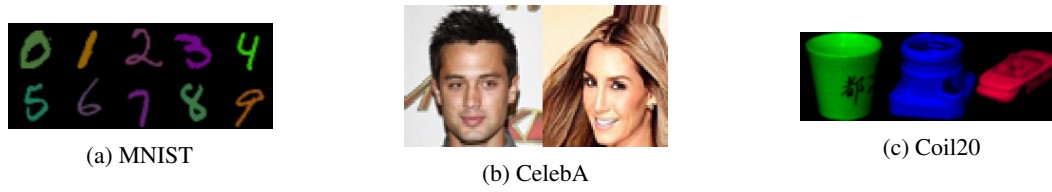

(a) MNIST           (b) CelebA           (c) Coil20

Figure 2: Datasets with spurious features. The object color spuriously correlates with the label in MNIST (a) and Coil20 (c) datasets. The hair color spuriously correlates with gender in the CelebA dataset (b).

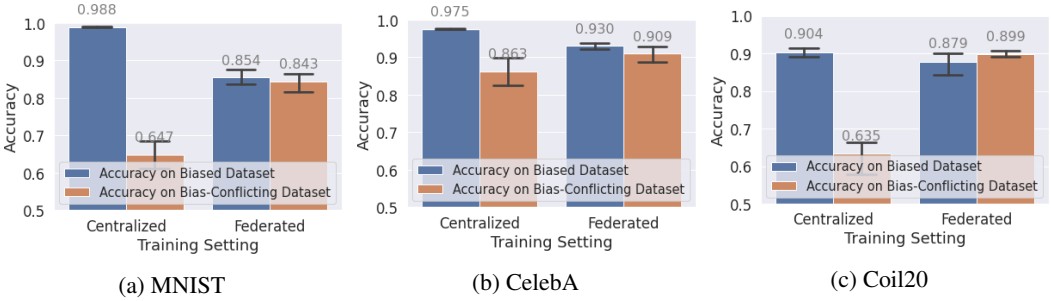

(a) MNIST           (b) CelebA           (c) Coil20

Figure 3: The accuracy of ML models on Biased Dataset and Bias-Conflicting Dataset under centralized and federated training settings. In the centralized setting, an ML model is trained by a single dataset that contains all the samples with a fixed spurious correlation. The global models in the federated setting achieve smaller accuracy disparities between biased and bias-conflicting datasets.

where $f$ is the victim ML model, and $\epsilon$ is the attack budget. In this example, we consider the $L_2$ attacks, but extensions to general $L_p$ attacks are straightforward. We say an adversarial example to be *transferable* if it also fools another ML model (e.g., a personalized model) other than the original victim model $f$ (e.g., the global model).

## 4   AN EMPIRICAL STUDY WITH SPURIOUS FEATURES

To gain some insights into the problem, we first perform an empirical study on the accuracy disparity of the global and personalized models in an FL setting. In this study, the personalization method is fine-tuning. Our results highlight the risk of existing fine-tuning-based personalization methods and the difficulty of mitigating the risk. We also highlight the correlation between the adversarial transferability and the accuracy disparity between the global and personalized models. We provide additional theoretical analysis in Section 5 to support the observed correlation. The spurious features in the empirical study are as follows.

**Spurious Features**   We consider color as the spurious feature for the MNIST, CelebA, and Coil20 datasets. In the MNIST and Coil20 datasets, we manually color the objects according to their labels to create spurious correlations, as Figure 2 shows. The spurious correlations vary across clients for the MNIST and Coil20 data (e.g., the red color correlates with label zero on the first client and with label one on the second client) to create additional statistical heterogeneity. In the CelebA dataset, the hair color attribute correlates with the gender label. We assign disjoint subsets of celebrities to different users, which naturally increases statistical heterogeneity for the spurious correlation.

### 4.1   STATISTICAL HETEROGENEITY REDUCES ACCURACY DISPARITY

Figure 3 shows the accuracy disparity of ML models on biased and bias-conflicting test sets. Compared to the models trained in the centralized setting, where the spurious correlations are fixed, the accuracy disparity of models trained in the federated setting decreased significantly. These empirical results suggest that the global model in FL is more robust to spurious features if the spurious features are non-i.i.d. across clients.

To explain this observation, consider the relationship between the gradient directions and the learned features. For the spurious features, its correlation with the label may change across clients. As

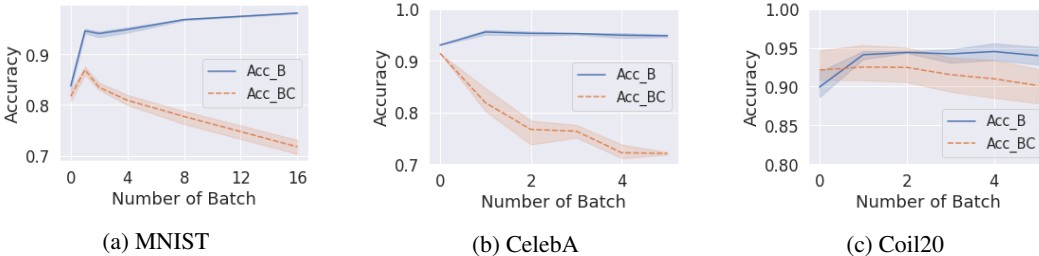

(a) MNIST  (b) CelebA  (c) Coil20

Figure 4: The accuracy of personalized model on Biased Dataset (Acc_B) and Bias-Conflicting Dataset (Acc_BC) with increasing fine-tuning batches. The personalized models entangle spurious features and increase accuracy disparities between biased and bias-conflicting datasets.

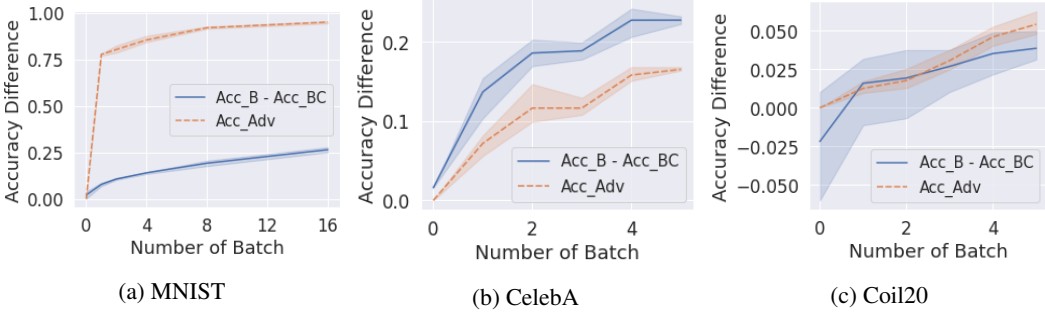

(a) MNIST  (b) CelebA  (c) Coil20

Figure 5: The accuracy disparity of personalized models on biased dataset and bias-conflicting dataset (Acc_B - Acc_BC) and their accuracy on adversarial examples (Acc_Adv). As the personalized models entangle spurious features and increase the accuracy disparity, the accuracy of the personalized models on adversarial examples increases, which indicates the adversarial transferability between the global and personalized models decreases.

an example, in some users' local datasets, the blond hair does not correlate with the gender label because the dataset does not contain any blond female image or the dataset has blond male images, as visualized in Figure 1a. Therefore, the gradient directions for the spurious features are diverse across clients, as shown in Figure 1b. The divergence between the gradient directions, as a consequence, makes learning spurious features difficult. In contrast, the non-spurious features, e.g., shape features, are more consistent across clients, leading to a more consistent gradient direction.

## 4.2 PERSONALIZATION MAY EXACERBATE ACCURACY DISPARITY

Although the global model in the federated setting has a lower accuracy disparity than in the centralized setting, the advantage could vanish during the personalization step.

**The Personalized Model Entangles Spurious Features** As can be observed in Figure 4, the accuracy first increases and decreases on the MNIST bias-conflicting test set and slowly decreases on Coil20. These two observations indicate that the personalized model entangles spurious features and exacerbates the accuracy disparity in a few batches. Although in principle, one may resort to early stopping, this is not feasible when the bias-conflicting dataset is unavailable or scarce.

## 4.3 ADVERSARIAL TRANSFERABILITY INDICATES ACCURACY DISPARITY

Because the bias-conflicting test set is often unavailable, it is infeasible to directly measure the accuracy disparity across personalized models. To this end, in this section, we focus on methods that implicitly measure the accuracy disparity and determine whether the personalized models entangle spurious features. Following our hypothesis in Section 1, we consider using the adversarial transferability between the global and personalized models as a proxy for the accuracy disparity measurement. Figure 5 plots the accuracy disparity and the adversarial transferability during fine-tuning. As the accuracy disparity of the personalized models increases and drifts away from that of the global model, the adversarial transferability between the global and personalized models decreases. This result empirically validates our hypothesis.

## 5 THEORETICAL INSIGHTS

This section presents our theoretical result that supports our hypothesis in Section 1 and the experimental results in Section 4.3. Our theoretical result applies the loss, which has similar behavior to the accuracy as is shown in the empirical results in Section 7. Before we proceed, some additional definitions and notations are needed for the presentation, and we provide a table summarizing all the notations used in Appendix A to ease the reading. Then, we connect both the loss disparity and the adversarial transferability to the angle between the gradients of the global and personalized models. In what follows, we shall show an upper bound of the loss disparity of a personalized model, which consists of the adversarial transferability between the global and personalized models and an indicator of the entanglement of the global model to spurious features.

### 5.1 MORE DEFINITIONS AND NOTATION

We define natural perturbation $\boldsymbol{\Delta}$ to model the distribution shift between the bias-conflicting $\mathcal{D}_{bc}$ and biased $\mathcal{D}_b$. $\boldsymbol{\Delta}$ could change a bias-aligned sample to a corresponding bias-conflicting sample. The distribution $\mathcal{D}_{\boldsymbol{\Delta}|\boldsymbol{x}}$ of the natural perturbation $\boldsymbol{\Delta}$ conditions on the data sample $\boldsymbol{x}$. Formally, for any $\boldsymbol{x} \in \mathcal{R}^d$, we have: $\mathrm{Pr}_{\boldsymbol{x} \sim \mathcal{D}_{bc}}(\boldsymbol{x}) = \sum_{\boldsymbol{x}' \in \mathcal{R}^d, \boldsymbol{\Delta} \in \mathcal{R}^d} \mathbf{1}_{\{\boldsymbol{x}=\boldsymbol{x}'+\boldsymbol{\Delta}\}} \cdot \mathrm{Pr}_{\boldsymbol{x}' \sim \mathcal{D}_b}(\boldsymbol{x}') \cdot \mathrm{Pr}_{\boldsymbol{\Delta} \sim \mathcal{D}_{\Delta}|\boldsymbol{x}'}(\boldsymbol{\Delta})$.

**Running Example** For a non-blond male image $\boldsymbol{x}$, we could draw a natural perturbation $\boldsymbol{\Delta}$ from $\mathcal{D}_{\boldsymbol{\Delta}|\boldsymbol{x}}$ that change the hair color in $\boldsymbol{x}$ to blond. That is saying, $\boldsymbol{x}+\boldsymbol{\Delta}$ is a blond male image. Iteratively drawing data samples from the biased dataset and applying the sampled natural perturbations to the data samples result in a dataset with bias-conflicting samples.

Another perturbation to consider is the adversarial perturbation $\boldsymbol{\delta}_{f,\epsilon} = \boldsymbol{x}_{adv} - \boldsymbol{x}$ that is generated using $f$ with budget $\epsilon$. Plugging the definition of $\boldsymbol{\delta}_{f,\epsilon}$ into Eq. (1), we have $\boldsymbol{\delta}_{f,\epsilon} = \arg\max_{\|\boldsymbol{\delta}\| \leq \epsilon} \ell(f(\boldsymbol{x} + \boldsymbol{\delta}), y)$. Since the budget $\epsilon$ is small, we could approximate the loss function $\ell$ using the first-order gradient: $\boldsymbol{\delta}_{f,\epsilon} = \arg\max_{\|\boldsymbol{\delta}\| \leq \epsilon} \nabla_{\boldsymbol{x}} \ell(f(\boldsymbol{x}), y)^{\top} \boldsymbol{\delta} = \epsilon \cdot \frac{\nabla_{\boldsymbol{x}} \ell(f(\boldsymbol{x}), y)}{\|\nabla_{\boldsymbol{x}} \ell(f(\boldsymbol{x}), y)\|}$ (Miyato et al., 2018; Liang et al., 2021). With the adversarial perturbation, we define the adversarial transferability loss:

$$\ell_{trans}(f_g, f_p, \boldsymbol{x}, y) = \Big(\ell(f_g(\boldsymbol{x} + \boldsymbol{\delta}_{f_g, \epsilon}), y) - \ell(f_g(\boldsymbol{x}), y)\Big) - \Big(\ell(f_p(\boldsymbol{x} + \boldsymbol{\delta}_{f_g, \epsilon}), y) - \ell(f_p(\boldsymbol{x}), y)\Big),$$

which indicates the effectiveness of the adversarial perturbation generated using the global model applied to the personalized models.

### 5.2 LOSS DISPARITY AND ADVERSARIAL TRANSFERABILITY

With the definitions of natural and adversarial perturbations, this section shows that both the loss disparity and the adversarial transferability connect to an angle $\theta$. Next, we outline the assumption:

**Assumption 1.** *The distribution shift does not exacerbate the entanglement of a model $f$ to spurious features $\boldsymbol{x}_s$, which is measured by $\nabla_{\boldsymbol{x}_s} \ell(f(\boldsymbol{x}), y)$:*

$$\mathbb{E}_{(\boldsymbol{x},y) \sim \mathcal{D}_b, \Delta \sim \mathcal{D}_{\Delta|\boldsymbol{x},y}} \Big[ \int_{\alpha=0}^{1} \langle \nabla_{\boldsymbol{x}_s} \ell(f(\boldsymbol{x} + \alpha \cdot \boldsymbol{\Delta}), y), \mathbf{1} \rangle \mathrm{d}\alpha \Big] \leq \mathbb{E}_{(\boldsymbol{x},y) \sim \mathcal{D}_b} [\langle \nabla_{\boldsymbol{x}_s} \ell(f(\boldsymbol{x}), y), \mathbf{1} \rangle].$$

Under Assumption 1, the following Lemmas hold.

**Lemma 1.** *Under Assumption 1, let $\Delta$ be the natural perturbation, $\theta$ be the angle between $\nabla_{\boldsymbol{x}} \ell(f_g(\boldsymbol{x}), y)$ and $\nabla_{\boldsymbol{x}} \ell(f_p(\boldsymbol{x}), y)$, $\theta_g$ be the angle between $\nabla_{\boldsymbol{x}} \ell(f_g(\boldsymbol{x}), y)$ and $\nabla_{\boldsymbol{x}_r} \ell(f_g(\boldsymbol{x}), y) \frown 0$, and $\gamma$ be $\frac{\|\nabla_{\boldsymbol{x}} \ell(f_g(\boldsymbol{x}), y)\|}{\|\nabla_{\boldsymbol{x}} \ell(f_p(\boldsymbol{x}), y)\|}$, we have:*

$$\begin{aligned}
\mathcal{L}(f_p, \mathcal{D}_{bc}) - \mathcal{L}(f_p, \mathcal{D}_b) &= \mathbb{E}_{(\boldsymbol{x},y) \sim \mathcal{D}_b, \boldsymbol{\Delta} \sim \mathcal{D}_{\boldsymbol{\Delta}|\boldsymbol{x}}} \Big[ \int_{\alpha=0}^{1} \langle \nabla_{\boldsymbol{x}_s} \ell(f_p(\boldsymbol{x} + \alpha \cdot \boldsymbol{\Delta}), y), \mathbf{1} \rangle \mathrm{d}\alpha \Big] \\
&< \mathbb{E}_{(\boldsymbol{x},y) \sim \mathcal{D}_b, \boldsymbol{\Delta} \sim \mathcal{D}_{\boldsymbol{\Delta}|\boldsymbol{x}}} \Big[ \frac{\sqrt{d_s}}{\gamma} \cdot \|\nabla_{\boldsymbol{x}} \ell(f_g(\boldsymbol{x}), y)\| \cdot (\sin\theta_g + \sin\theta) \Big]
\end{aligned} \tag{2}$$

Lemma 1 connects the loss disparity to $\theta$. The $\theta_g$, differing from $\theta$, is an indicator of the entanglement of the global model to spurious features and is a constant during the personalization step.

**Lemma 2.** *Let $\epsilon$ be the attack budget, $\theta$ be the angle between $\nabla_{\boldsymbol{x}}\ell(f_g(\boldsymbol{x}), y)$ and $\nabla_{\boldsymbol{x}}\ell(f_p(\boldsymbol{x}), y)$, $\gamma$ be $\frac{\|\nabla_{\boldsymbol{x}}\ell(f_g(\boldsymbol{x}), y)\|}{\|\nabla_{\boldsymbol{x}}\ell(f_p(\boldsymbol{x}), y)\|}$, and the loss function $\ell : \mathcal{Y} \times \mathcal{Y} \to \mathbb{R}$ be $\lambda$-smooth, twice differentiable, we have*

$$\epsilon \cdot \|\nabla_{\boldsymbol{x}}\ell(f_g(\boldsymbol{x}), y)\| \cdot (1 - \frac{1}{\gamma} \cdot \cos\theta) - \lambda \cdot \epsilon^2 \leq \ell_{trans}(f_g, f_p, \boldsymbol{x}, y)$$

$$\leq \epsilon \cdot \|\nabla_{\boldsymbol{x}}\ell(f_g(\boldsymbol{x}), y)\| \cdot (1 - \frac{1}{\gamma} \cdot \cos\theta) + \lambda \cdot \epsilon^2$$

(3)

Lemma 2 connects the adversarial transferability loss to $\theta$. In the following analysis, we connect the loss disparity to adversarial transferability via $\theta$.

### 5.3 A GENERALIZATION UPPER BOUND

We now present an upper bound of the disparity $\mathcal{L}(f_p, \mathcal{D}_{bc}) - \mathcal{L}(f_p, \mathcal{D}_b)$. The main idea is to derive an upper bound of $\|\nabla_{\boldsymbol{x}}\ell(f_g(\boldsymbol{x}), y)\| \cdot \sin\theta$ in Eq. (2) from Eq. (3).

**Theorem 3.** *Let $\gamma_{\min}$ be the minimum of $\gamma$, with Lemmas 1-2, we have:*

$$\mathcal{L}(f_p, \mathcal{D}_{bc}) - \mathcal{L}(f_p, \mathcal{D}_b) < \sqrt{d_s} \cdot \Big( (\frac{\sin\theta_g + \sqrt{2}}{\gamma_{\min}} - 1) \cdot \mathbb{E}_{(\boldsymbol{x},y)\sim\mathcal{D}_b}[\|\nabla_{\boldsymbol{x}}\ell(f_g(\boldsymbol{x}), y)\|]$$

$$+ \frac{1}{\epsilon} \cdot \mathbb{E}_{(\boldsymbol{x},y)\sim\mathcal{D}_b}[\ell_{trans}(f_g, f_p, \boldsymbol{x}, y)] + \lambda \cdot \epsilon \Big)$$

Theorem 3 suggests (1) debiasing the global model $f_g$, whose entanglement to spurious features is measured by $\theta_g$, and (2) enforcing the adversarial transferability between $f_g$ and $f_p$ help reducing the loss disparity of personalized models. For the constants $\gamma_{\min}$ and $\lambda$, we further explore their impacts in the following section and Appendix D.1, respectively.

## 6 METHODS

With the empirical and theoretical results in Sections 4.3 and 5, respectively, it is natural to ask if enforcing the adversarial transferability in the personalization step reduces the accuracy disparity. In this section, we first introduce adversarial examples to the personalization step as a regularization term added to the original loss function, aiming to *enforce the adversarial transferability*. However, the accuracy disparity still increases, albeit much slower, even if the adversarial transferability remains high. One possible reason is that the personalized model increases its gradient norm, which helps preserve the adversarial transferability but does not prevent the personalized model from entangling spurious features. To this end, we add an $L_2$ regularization term to the loss function, *aligning the gradient norms of the global and personalized models*. Combing these two methods addresses the accuracy disparity. Both methods are relatively light-weight from a computational perspective.

### 6.1 ENFORCING ADVERSARIAL TRANSFERABILITY

We enforce that global and personalized models make consistent predictions on adversarial examples, such that adversarial examples transfer from one to the other.

**Generating Adversarial Examples** The projected gradient descent (PGD) attack (Madry et al., 2018) is an effective attack method that uses the neural network's first-order gradient, and is easy to compute. Additionally, . The attack solves $\boldsymbol{x}_{adv} = \arg\max_{\|\boldsymbol{x}'-\boldsymbol{x}\|\leq\epsilon} \ell(f(\boldsymbol{x}'), y)$ iteratively. At iteration $t+1$, the adversarial example is: $\boldsymbol{x}_{adv}^{t+1} = \text{Proj}_{\|\boldsymbol{x}_{adv}-\boldsymbol{x}\|\leq\epsilon}(\boldsymbol{x} + \alpha \cdot \text{sign}(\nabla_{\boldsymbol{x}_{adv}^t}\ell(f_g(\boldsymbol{x}_{adv}^t), y)))$, where $\text{Proj}$ is a projection operator.

**Enforcing Consistent Predictions** Both the global model $f_g$ and the personalized model $f_p$ take the adversarial example $\boldsymbol{x}_{adv}$ as input and output $\boldsymbol{z}_g$ and $\boldsymbol{z}_p$ from their last layers, respectively. We enforce the adversarial transferability by adding the following regularization term, which maximizes the cross-entropy between $\boldsymbol{z}_g$ and $\boldsymbol{z}_p$. Since the global model $f_g$ is fixed as a reference in the personalization step and its low accuracy disparity is desirable, we use $\boldsymbol{z}_g$ as the ground-truth:

$$R_{adv}(\boldsymbol{z}_g, \boldsymbol{z}_p) = \sum_{i=1}^{K}[z_{g_i} \cdot \log(z_{p_i}) + (1 - z_{g_i}) \cdot \log(1 - z_{p_i})],$$

where $K$ is the number of classes. The local model has access to the global model, so there is no additional communication overhead for implementing this regularization. The adversarial examples are computed using the global model once for all. The computation only needs a few back-propagations, much less than training the global model.

## 6.2 Aligning Gradient Norms

In Eq. (3), we have seen that the adversarial transferability loss depends not only on the angle $\theta$, which connects the transferability to the disparity but also on $\gamma := \frac{\|\nabla_{\boldsymbol{x}}\ell(f_g(\boldsymbol{x}),y)\|}{\|\nabla_{\boldsymbol{x}}\ell(f_p(\boldsymbol{x}),y)\|}$. A small $\gamma$ indicates that the personalized model increases its gradient norm. Then, the personalized model could entangle the spurious features and increase $\theta$ without decreasing the adversarial transferability.

To prevent $\gamma$ from decreasing, we employ a simple and effective strategy by adding an $L_2$ regularization term $R_{L_2} = \|\boldsymbol{w}_g - \boldsymbol{w}_p\|^2$ to the loss function. The motivation behind the $L_2$ term is straightforward: if two models have similar weights, they have similar gradients. Empirical results in Appendix D.2 show the effectiveness of the $L_2$ term in controlling $\gamma$. Although prior works (Li et al., 2020; T. Dinh et al., 2020; Li et al., 2021) have explored similar regularization methods, we develop the regularization term from a different perspective.

## 7 Experiments

This section presents our experimental results, demonstrating that our method reduces the accuracy disparity. We also show that the benefit of enhanced average accuracy from fine-tuning is preserved.

## 7.1 Setting

**Data Partition** We distribute the MNIST and Coil20 dataset across 50 clients where each client have 5 different classes. The local dataset on each client is further partitioned to train/validation/test set with a ratio of 72:8:20, following prior work (Li et al., 2021). We make two data partitions for CelebA: CelebA_R using a real partition and CelebA_S using a synthetic partition, both have 508 clients. In both CelebA partitions, each client represent a disjoint set of celebrity (Li et al., 2021).

Due to the limited space, we further detail the data partition in Appendix C.1, report the hyper-parameters in Appendix C.2, and list the neural network architecture in Appendix C.3.

## 7.2 The Effectiveness of Proposed Methods

We conduct an ablation study on the MNIST dataset. Figures 6b and 6f demonstrate the effectiveness of enforcing adversarial transferability. However, the accuracy and loss disparity still increase during personalization, which is potentially caused by the gradient norm issue (Section 6.2). Aligning the gradient norms by applying the $L_2$ regularization term while enforcing adversarial transferability address the accuracy and loss disparity as Figures 6d and 6h show, respectively. Figure 6c and 6g further show that applying the $L_2$ regularization term alone does not address the accuracy or loss disparity. Compared to naive fine-tuning, which is reported in Figures 6a and 6e, our method mitigates the accuracy and loss disparities by $\sim 50\%$.

## 7.3 Analysis

We compare our method to no personalization (Global), naive fine-tuning (FT), Ditto (Li et al., 2021), up-weighting (UW) (Sagawa et al., 2020), and just train twice (JTT) (Liu et al., 2021). The up-weighting method is implemented via sampling bias-aligned and bias-conflicting samples with equal probability (Sagawa et al., 2020). Up-weighting and JTT are not applicable to Coil20 due to the lack of bias-conflicting samples. We use the local finetuning version of the Ditto solver because the local finetuning solver performs fewer local updates than that of the joint optimization solver and therefore entangles spurious features less. Each experiment is repeated 9 times with 3 random seeds for the federated learning step and 3 for the personalization step. We select models using the validation accuracy minus the decrease of adversarial transferability and using the validation accuracy for other baseline and competitor methods.

Tables 1 shows the main result. Our method reduces the accuracy disparity of personalized models from $15.12\%$ to $2.15\%$, compared to other personalization methods. Our method also preserves the

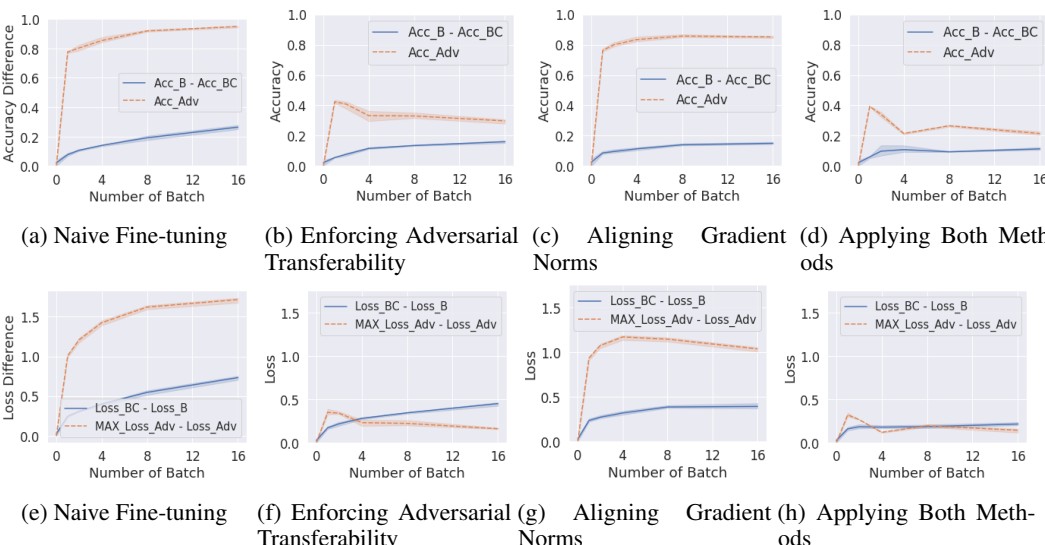

Figure 6: The accuracy disparity (Acc_B - Acc_BC) and adversarial transferability accuracy (Acc_Adv), and the loss disparity (Loss_BC - Loss_B) and adversarial transferability loss (MAX_Loss_Adv - Loss_Adv) with different methods. Combining the two proposed methods addresses the accuracy and loss disparities. Acc_BC/Loss_BC and Acc_B/Loss_B are the accuracy/losses on the bias-conflicting and biased test sets, respectively. Acc_Adv/Loss_Adv is the accuracy/loss on adversarial examples. Acc_Adv and MAX_Loss_Adv - Loss_Adv, which measures the decrease of Loss_Adv, indicate the decrease of adversarial transferability.

Table 1: Accuracy of personalized Models on Biased Test Set (Acc_B) and Bias-Conflicting Test Set (Acc_B). Our proposed method achieves the lowest accuracy disparity (2.15%) compared to other personalization methods (15.12%/15.38%), and $3.43\%$ accuracy improvement on the biased test set and $0.85\%$ improvement on the biased-conflicting test set compared to the global model.

| Method | MNIST | | CelebA_S | | CelebA_R | | Coil20 | |
| | Acc_B | Acc_BC | Acc_B | Acc_BC | Acc_B | Acc_BC | Acc_B | Acc_BC |
| --- | --- | --- | --- | --- | --- | --- | --- | --- |
| Global | $.852_{\pm 2e-4}$ | $.847_{\pm 6e-4}$ | $.930_{\pm 5e-5}$ | $.910_{\pm 3e-4}$ | $.909_{\pm 6e-5}$ | $.929_{\pm 5e-5}$ | $.882_{\pm .6e-4}$ | $.903_{\pm 7e-4}$ |
| FT | $.989_{\pm 6e-7}$ | $.704_{\pm 3e-4}$ | $.952_{\pm 6e-5}$ | $.786_{\pm 5e-4}$ | $.963_{\pm 6e-6}$ | $.849_{\pm 1e-3}$ | $.931_{\pm 1e-4}$ | $.891_{\pm 2e-4}$ |
| Ditto | $.982_{\pm 3e-6}$ | $.724_{\pm 1e-3}$ | $.948_{\pm 5e-5}$ | $.715_{\pm 5e-4}$ | $.966_{\pm 1e-5}$ | $.884_{\pm 2e-4}$ | $.939_{\pm 4e-5}$ | $.897_{\pm 3e-4}$ |
| UW | $.968_{\pm 2e-5}$ | $.823_{\pm 7e-4}$ | $.930_{\pm 7e-6}$ | $.889_{\pm 5e-4}$ | $.936_{\pm 1e-5}$ | $.895_{\pm 4e-4}$ | N/A | N/A |
| JTT | $.985_{\pm 2e-7}$ | $.707_{\pm 6e-5}$ | $.952_{\pm 2e-6}$ | $.817_{\pm 3e-4}$ | $.956_{\pm 2e-5}$ | $.836_{\pm 1e-3}$ | N/A | N/A |
| Ours | $.951_{\pm 2e-5}$ | $.870_{\pm 8e-4}$ | $.932_{\pm 3e-5}$ | $.910_{\pm 2e-4}$ | $.925_{\pm 2e-5}$ | $.927_{\pm 8e-5}$ | $.901_{\pm 3e-4}$ | $.916_{\pm 5e-8}$ |

enhanced average accuracy from fine-tuning, resulting in $3.43\%$ accuracy improvement on the biased test set and $0.85\%$ improvement on the biased-conflicting test set compared to the global model. In contrast, the naive fine-tuning method sacrifices the accuracy on the bias-conflicting test set by up to $14.3\%$ and increase the accuracy disparity by $15.12\%$. We also find that our methods outperform the supervised up-weighting method and the unsupervised JTT method, which increase the average accuracy disparity to $7.56\%$ and $17.76\%$, respectively. One possible reason is that the diversity of the up-weighted bias-conflicting samples are small. Therefore, the neural network could memorize them instead of discarding spurious features. Appendix D.3 further shows empirical results that support our analysis.

## 8 CONCLUSION

In this work, we show the risk of prior federated learning personalization methods with spurious features, which lead to high accuracy disparity between the global and local models. Then, we develop a strategy by enforcing the adversarial transferability between the global and personalized models to reduce the accuracy disparity. Both empirical and theoretical results show that our strategy is effective.

## 9    ETHICS STATEMENT

Our method mitigates the issue of spurious features, which lead to bias towards minority groups, in personalized federated learning. However, completely disentangling spurious features remains challenging and is an issue for many federated learning methods.

## 10    REPRODUCIBILITY STATEMENT

Our implementation, including data partition scripts, is based on the FedML library (He et al., 2020), which is open-sourced. The proofs of the theoretical results are in Appendix B. The datasets in our experiments are publicly available. We detail the data partition in Appendix C.1, report the hyper-parameter tuning in Appendix C.2, and list the neural network architecture in Appendix C.3.

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

# Appendix

## A    NOTATION

Table 2: Table of Notation

| Symbol | Description |
|---|---|
| $\boldsymbol{x}, y$ | A pair of data sample and label |
| $\boldsymbol{x}_r, \boldsymbol{x}_s$ | The robust feature and spurious features in $\boldsymbol{x} = [\boldsymbol{x}_r, \boldsymbol{x}_s]$, respectively |
| $d, d_r, d_s$ | The dimension of $\boldsymbol{x}, \boldsymbol{x}_r, \boldsymbol{x}_s$, respectively |
| $f_g$ | The global model |
| $f_p$ | The personlized local model |
| $\boldsymbol{\delta}_{f_g, \epsilon}$ | An adversarial purtubation generated by the global model $f_g$ with attack budget $\epsilon$ |
| $\boldsymbol{\Delta}$ | An natural perturbation, which could flip the spurious attribute |
| $\mathcal{D}_g$ | The global distribution, which is the union of local distributions |
| $\mathcal{D}_b$ | A biased local distribution |
| $\mathcal{D}_{bc}$ | A bias-conflicting local distribution |
| $\mathcal{D}_{\boldsymbol{\Delta} \mid \boldsymbol{x}, y}$ | The distribution of natural perturbation |
| $\mathrm{supp}(\mathcal{D})$ | The support of distribution $\mathcal{D}$ |
| $\langle \cdot, \cdot \rangle$ | An inner product of two vectors |
| $\cdot \frown \cdot$ | A concatenation of two vectors |

## B    PROOFS

### B.1    PROOF OF LEMMA 1

**Lemma 1.** *Let $\Delta$ be the natural perturbation, $\theta$ be the angle between $\nabla_{\boldsymbol{x}}\ell(f_g(\boldsymbol{x}), y)$ and $\nabla_{\boldsymbol{x}}\ell(f_p(\boldsymbol{x}), y)$, $\theta_g$ be the angle between $\nabla_{\boldsymbol{x}}\ell(f_g(\boldsymbol{x}), y)$ and $\nabla_{\boldsymbol{x}_r}\ell(f_g(\boldsymbol{x}), y) \frown \mathbf{0}$, and $\gamma$ be $\frac{\|\nabla_{\boldsymbol{x}}\ell(f_g(\boldsymbol{x}), y)\|}{\|\nabla_{\boldsymbol{x}}\ell(f_p(\boldsymbol{x}), y)\|}$, we have:*

$$\mathcal{L}(f_p, \mathcal{D}_{bc}) - \mathcal{L}(f_p, \mathcal{D}_b) = \mathbb{E}_{(\boldsymbol{x}, y) \sim \mathcal{D}_b, \boldsymbol{\Delta} \sim \mathcal{D}_{\boldsymbol{\Delta} \mid \boldsymbol{x}}} [\int_{\alpha=0}^{1} \langle \nabla_{\boldsymbol{x}_s} \ell(f_p(\boldsymbol{x} + \alpha \cdot \boldsymbol{\Delta}), y), \mathbf{1} \rangle \mathrm{d}\alpha]$$

$$< \mathbb{E}_{(\boldsymbol{x}, y) \sim \mathcal{D}_b} [\frac{\sqrt{d_s}}{\gamma} \cdot \|\nabla_{\boldsymbol{x}} \ell(f_g(\boldsymbol{x}), y)\| \cdot (\sin\theta_g + \sin\theta)]$$

*Proof.* Rewriting $\mathcal{L}(f_p, \mathcal{D}_{bc})$ and introducing $\boldsymbol{\Delta}$:

$$\begin{aligned}
\mathcal{L}(f_p, \mathcal{D}_{bc}) &= \mathbb{E}_{(\boldsymbol{x}, y) \sim \mathcal{D}_{bc}} [\ell(f_p(\boldsymbol{x}), y)] \\
&= \mathbb{E}_{(\boldsymbol{x}, y) \sim \mathcal{D}_b, \boldsymbol{\Delta} \sim \mathcal{D}_{\boldsymbol{\Delta} \mid \boldsymbol{x}}} [\ell(f_p(\boldsymbol{x} + \boldsymbol{\Delta}), y)] \\
&= \mathbb{E}_{(\boldsymbol{x}, y) \sim \mathcal{D}_b, \boldsymbol{\Delta} \sim \mathcal{D}_{\boldsymbol{\Delta} \mid \boldsymbol{x}}} [\ell(f_p(\boldsymbol{x}), y) + \ell(f_p(\boldsymbol{x} + \boldsymbol{\Delta}), y) - \ell(f_p(\boldsymbol{x}), y)] \\
&= \mathbb{E}_{(\boldsymbol{x}, y) \sim \mathcal{D}_b, \boldsymbol{\Delta} \sim \mathcal{D}_{\boldsymbol{\Delta} \mid \boldsymbol{x}}} [\ell(f_p(\boldsymbol{x}), y)] \\
&\quad + \mathbb{E}_{(\boldsymbol{x}, y) \sim \mathcal{D}_b, \boldsymbol{\Delta} \sim \mathcal{D}_{\boldsymbol{\Delta} \mid \boldsymbol{x}}} [\ell(f_p(\boldsymbol{x} + \boldsymbol{\Delta}), y) - \ell(f_p(\boldsymbol{x}), y)] \\
&= \mathbb{E}_{(\boldsymbol{x}, y) \sim \mathcal{D}_b} [\ell(f_p(\boldsymbol{x}), y)] \\
&\quad + \mathbb{E}_{(\boldsymbol{x}, y) \sim \mathcal{D}_b, \boldsymbol{\Delta} \sim \mathcal{D}_{\boldsymbol{\Delta} \mid \boldsymbol{x}}} [\ell(f_p(\boldsymbol{x} + \boldsymbol{\Delta}), y) - \ell(f_p(\boldsymbol{x}), y)] \\
&= \mathcal{L}(f_p, \mathcal{D}_b) + \mathbb{E}_{(\boldsymbol{x}, y) \sim \mathcal{D}_b, \boldsymbol{\Delta} \sim \mathcal{D}_{\boldsymbol{\Delta} \mid \boldsymbol{x}}} [\int_{\alpha=0}^{1} \langle \nabla_{\boldsymbol{x}} \ell(f_p(\boldsymbol{x} + \alpha \cdot \boldsymbol{\Delta}), y), \mathbf{1} \rangle \mathrm{d}\alpha] \\
&= \mathcal{L}(f_p, \mathcal{D}_b) + \mathbb{E}_{(\boldsymbol{x}, y) \sim \mathcal{D}_b, \boldsymbol{\Delta} \sim \mathcal{D}_{\boldsymbol{\Delta} \mid \boldsymbol{x}}} [\int_{\alpha=0}^{1} \langle \nabla_{\boldsymbol{x}_s} \ell(f_p(\boldsymbol{x} + \alpha \cdot \boldsymbol{\Delta}), y), \mathbf{1} \rangle \mathrm{d}\alpha]
\end{aligned}$$

Moving $\mathcal{L}(f_p, \mathcal{D}_b)$ to the left-hand-side (LHS), we have:

$$\mathcal{L}(f_p, \mathcal{D}_{bc}) - \mathcal{L}(f_p, \mathcal{D}_b) = \mathbb{E}_{(\boldsymbol{x},y) \sim \mathcal{D}_b, \boldsymbol{\Delta} \sim \mathcal{D}_{\boldsymbol{\Delta}|\boldsymbol{x}}} \left[ \int_{\alpha=0}^{1} \langle \nabla_{\boldsymbol{x}_s} \ell(f_p(\boldsymbol{x} + \alpha \cdot \boldsymbol{\Delta}), y), \mathbf{1} \rangle \mathrm{d}\alpha \right]. \quad (4)$$

According to Assumption 1, we further have:

$$\mathbb{E}_{(\boldsymbol{x},y) \sim \mathcal{D}_b, \boldsymbol{\Delta} \sim \mathcal{D}_{\boldsymbol{\Delta}|\boldsymbol{x}}} \left[ \int_{\alpha=0}^{1} \langle \nabla_{\boldsymbol{x}_s} \ell(f_p(\boldsymbol{x} + \alpha \cdot \boldsymbol{\Delta}), y), \mathbf{1} \rangle \right] \leq \mathbb{E}_{\boldsymbol{x} \sim \mathcal{D}_b} [\langle \nabla_{\boldsymbol{x}_s} \ell(f_p(\boldsymbol{x}), y), \mathbf{1} \rangle] \quad (5)$$

Next, we connect $\langle \nabla_{\boldsymbol{x}_s} \ell(f_p(\boldsymbol{x} + \alpha \cdot \boldsymbol{\Delta}), y), \mathbf{1} \rangle$ to $\|\nabla_{\boldsymbol{x}} \ell(f_g(\boldsymbol{x}), y)\| \cdot \sin\theta$. The first step is connecting $\langle \nabla_{\boldsymbol{x}_s} \ell(f_p(\boldsymbol{x} + \alpha \cdot \boldsymbol{\Delta}), y), \mathbf{1} \rangle$ to $\|\nabla_{\boldsymbol{x}_s} \ell(f_p(\boldsymbol{x}), y)\|$ using Cauchy-Schwarz inequality:

$$\begin{aligned}
\langle \nabla_{\boldsymbol{x}_s} &\ell(f_p(\boldsymbol{x} + \alpha \cdot \boldsymbol{\Delta}), y), \mathbf{1} \rangle \\
&\leq \sqrt{\langle \nabla_{\boldsymbol{x}_s} \ell(f_p(\boldsymbol{x} + \alpha \cdot \boldsymbol{\Delta}), y), \nabla_{\boldsymbol{x}_s} \ell(f_p(\boldsymbol{x} + \alpha \cdot \boldsymbol{\Delta}), y) \rangle \cdot \langle \mathbf{1}, \mathbf{1} \rangle} \\
&= \sqrt{d_s} \cdot \|\nabla_{\boldsymbol{x}_s} \ell(f_p(\boldsymbol{x}), y)\|
\end{aligned} \quad (6)$$

Then, we connect $\|\nabla_{\boldsymbol{x}_s} \ell(f_p(\boldsymbol{x}), y)\|$ to $\|\nabla_{\boldsymbol{x}} \ell(f_g(\boldsymbol{x}), y)\|$. Assuming the global model $f_g$ entangles spurious features and the angle between $\nabla_{\boldsymbol{x}} \ell(f_g(\boldsymbol{x}), y)$ and $\nabla_{\boldsymbol{x}_r} \ell(f_g(\boldsymbol{x}), y) \frown \mathbf{0}$ is $\theta_g$, we have:

$$\|\nabla_{\boldsymbol{x}_s} \ell(f_p(\boldsymbol{x}), y)\| \leq \|\nabla_{\boldsymbol{x}} \ell(f_g(\boldsymbol{x}), y)\| \cdot \sin(\theta_g + \theta). \quad (7)$$

Since it is easy to see that $\theta \in [0, \frac{\pi}{4}]$ and the gradient of $\sin\theta$ is monotonically decreasing in $[0, \frac{\pi}{4}]$, we have:

$$\begin{aligned}
\sin(\theta_g + \theta) &= \int_0^{\theta_g + \theta} \nabla \sin\theta \mathrm{d}\theta \\
&= \int_0^{\theta_g + \theta} \cos\theta \mathrm{d}\theta \\
&< \int_0^{\theta_g} \cos\theta \mathrm{d}\theta + \int_0^{\theta} \cos\theta \mathrm{d}\theta \\
&= \sin\theta_g + \sin\theta
\end{aligned} \quad (8)$$

Combining Eq. (7) and Eq. (8), we have:

$$\|\nabla_{\boldsymbol{x}_s} \ell(f_p(\boldsymbol{x}), y)\| < \|\nabla_{\boldsymbol{x}} \ell(f_p(\boldsymbol{x}), y)\| \cdot (\sin\theta_g + \sin\theta) \quad (9)$$

Recalling the defition of $\gamma := \frac{\|\nabla_{\boldsymbol{x}} \ell(f_g(\boldsymbol{x}), y)\|}{\|\nabla_{\boldsymbol{x}} \ell(f_p(\boldsymbol{x}), y)\|}$ and combining Eq. (4), Eq. (5), Eq. (6), Eq. (9) complete the proof.

$\square$

## B.2 PROOF OF LEMMA 2

**Lemma 2.** *Let $\epsilon$ be the attack budget, $\theta$ be the angle between $\nabla_{\boldsymbol{x}} \ell(f_g(\boldsymbol{x}), y)$ and $\nabla_{\boldsymbol{x}} \ell(f_p(\boldsymbol{x}), y)$, $\gamma$ be $\frac{\|\nabla_{\boldsymbol{x}} \ell(f_g(\boldsymbol{x}), y)\|}{\|\nabla_{\boldsymbol{x}} \ell(f_p(\boldsymbol{x}), y)\|}$, and the loss function $\ell : \mathcal{Y} \times \mathcal{Y} \to \mathbb{R}$ be $\lambda$-smooth, twice differentiable, we have*

$$\epsilon \cdot \|\nabla_{\boldsymbol{x}} \ell(f_g(\boldsymbol{x}), y)\| \cdot (1 - \frac{1}{\gamma} \cdot \cos\theta) - \lambda \cdot \epsilon^2 \leq \ell_{trans}(f_g, f_p, \boldsymbol{x}, y)$$

$$\leq \epsilon \cdot \|\nabla_{\boldsymbol{x}} \ell(f_g(\boldsymbol{x}), y)\| \cdot (1 - \frac{1}{\gamma} \cdot \cos\theta) + \lambda \cdot \epsilon^2$$

*Proof.* Under the definition of the adversarial perturbation, it is easy to see that $\boldsymbol{\delta}_{f,\epsilon} = \epsilon \cdot \frac{\nabla_{\boldsymbol{x}} \ell(f(\boldsymbol{x}),y)}{\|\nabla_{\boldsymbol{x}} \ell(f(\boldsymbol{x}),y)\|}$ and $\boldsymbol{\delta}_{f,\epsilon}$ increases the loss by:

$$\ell(f_g(\boldsymbol{x} + \boldsymbol{\delta}_{f_g,\epsilon}), y) - \ell(f_g(\boldsymbol{x}), y) = \boldsymbol{\delta}_{f_g,\epsilon} \nabla_{\boldsymbol{x}} \ell(f_g(\boldsymbol{x}), y) + \frac{1}{2} \boldsymbol{\delta}_{f_g,\epsilon}^\top \nabla_{\tilde{\boldsymbol{x}}_g}^2 \ell(f_g(\tilde{\boldsymbol{x}}_g), y) \boldsymbol{\delta}_{f_g,\epsilon}$$

$$= \epsilon \cdot \|\nabla_{\boldsymbol{x}} \ell(f_g(\boldsymbol{x}), y)\| + \frac{1}{2} \boldsymbol{\delta}_{f_g,\epsilon}^\top \nabla_{\tilde{\boldsymbol{x}}_g}^2 \ell(f_g(\tilde{\boldsymbol{x}}_g), y) \boldsymbol{\delta}_{f_g,\epsilon}$$

where $\tilde{\boldsymbol{x}}_g$ is a linear interpolation between $\boldsymbol{x}$ and $\boldsymbol{x} + \boldsymbol{\delta}_{f_g,\epsilon}$, by the Lagrange's mean-value theorem. Similarly, for a transferable adversarial example from $f_g$ applies to $f_p$, $\boldsymbol{\delta}_{f_g,\epsilon}$ could increase the loss of $f_p$ by:

$$\ell(f_p(\boldsymbol{x} + \boldsymbol{\delta}_{f_g,\epsilon}), y) - \ell(f_p(\boldsymbol{x}), y) = \boldsymbol{\delta}_{f_g,\epsilon} \nabla_{\boldsymbol{x}} \ell(f_p(\boldsymbol{x}), y) + \frac{1}{2} \boldsymbol{\delta}_{f_g,\epsilon}^\top \nabla_{\tilde{\boldsymbol{x}}_p}^2 \ell(f_p(\tilde{\boldsymbol{x}}_p), y) \boldsymbol{\delta}_{f_g,\epsilon}$$

$$= \epsilon \cdot \|\nabla_{\boldsymbol{x}} \ell(f_p(\boldsymbol{x}), y)\| \cdot \cos\theta + \frac{1}{2} \boldsymbol{\delta}_{f_g,\epsilon}^\top \nabla_{\tilde{\boldsymbol{x}}_p}^2 \ell(f_p(\tilde{\boldsymbol{x}}_p), y) \boldsymbol{\delta}_{f_g,\epsilon}$$

where $\cos\theta = \frac{\nabla_{\boldsymbol{x}} \ell(f_g(\boldsymbol{x}),y) \cdot \nabla_{\boldsymbol{x}} \ell(f_p(\boldsymbol{x}),y)}{\|\nabla_{\boldsymbol{x}} \ell(f_g(\boldsymbol{x}),y)\| \|\nabla_{\boldsymbol{x}} \ell(f_p(\boldsymbol{x}),y)\|}$. Plugging the approximations above to the adversarial transferability loss, we have:

$$\ell_{trans}(f_g, f_p, \boldsymbol{x}, y) = \Big( \ell(f_g(\boldsymbol{x} + \boldsymbol{\delta}_{f_g,\epsilon}), y) - \ell(f_g(\boldsymbol{x}), y) \Big) - \Big( \ell(f_p(\boldsymbol{x} + \boldsymbol{\delta}_{f_g,\epsilon}), y) - \ell(f_p(\boldsymbol{x}), y) \Big)$$

$$= \epsilon \cdot \|\nabla_{\boldsymbol{x}} \ell(f_g(\boldsymbol{x}), y)\| - \epsilon \cdot \|\nabla_{\boldsymbol{x}} \ell(f_p(\boldsymbol{x}), y)\| \cdot \cos\theta$$

$$+ \frac{1}{2} \cdot \boldsymbol{\delta}_{f_g,\epsilon}^\top \nabla_{\tilde{\boldsymbol{x}}_g}^2 \ell(f_g(\tilde{\boldsymbol{x}}_g), y) \boldsymbol{\delta}_{f_g,\epsilon} - \frac{1}{2} \cdot \boldsymbol{\delta}_{f_g,\epsilon}^\top \nabla_{\tilde{\boldsymbol{x}}_p}^2 \ell(f_p(\tilde{\boldsymbol{x}}_p), y) \boldsymbol{\delta}_{f_g,\epsilon}$$

Under the $\lambda$-smooth assumption on the loss function, the spectral norms of the Hessian metrics are bounded. Therefore, we could bound the norm of the deviate between the quadratic terms (Nesterov, 2003, Proof of Theorem 2.1.5) in the adversarial transferability loss:

$$\|\boldsymbol{\delta}_{f_g,\epsilon}^\top \nabla_{\tilde{\boldsymbol{x}}_g}^2 \ell(f_g(\tilde{\boldsymbol{x}}_g), y) \boldsymbol{\delta}_{f_g,\epsilon} - \boldsymbol{\delta}_{f_g,\epsilon}^\top \nabla_{\tilde{\boldsymbol{x}}_p}^2 \ell(f_p(\tilde{\boldsymbol{x}}_p), y) \boldsymbol{\delta}_{f_g,\epsilon}\| \leq 2\lambda \cdot \boldsymbol{\delta}_{f_g,\epsilon}^\top \boldsymbol{\delta}_{f_g,\epsilon} = 2\lambda \cdot \epsilon^2 \quad (10)$$

Since the quadratic terms in Eq. (10) are scalars, we have:

$$-2\lambda \cdot \epsilon^2 \leq \boldsymbol{\delta}_{f_g,\epsilon}^\top \nabla_{\tilde{\boldsymbol{x}}_g}^2 \ell(f_g(\tilde{\boldsymbol{x}}_g), y) \boldsymbol{\delta}_{f_g,\epsilon} - \boldsymbol{\delta}_{f_g,\epsilon}^\top \nabla_{\tilde{\boldsymbol{x}}_p}^2 \ell(f_p(\tilde{\boldsymbol{x}}_p), y) \boldsymbol{\delta}_{f_g,\epsilon} \leq 2\lambda \cdot \epsilon^2 \quad (11)$$

Plugging Eq. (11) and the definition of $\gamma$ to $\ell_{trans}(f_g, f_p, \boldsymbol{x}, y)$ completes the proof.

$\square$

### B.3 Proof of Theorem 3

**Theorem 3.** *Let $\gamma_{\min}$ be the minimum of $\gamma$, under Assumptions 1 and Lemmas 1-2, we have:*

$$\mathcal{L}(f_p, \mathcal{D}_{bc}) - \mathcal{L}(f_p, \mathcal{D}_b) < \sqrt{d_s} \cdot \Big( (\frac{\sin\theta_g + \sqrt{2}}{\gamma_{\min}} - 1) \cdot \mathbb{E}_{(\boldsymbol{x},y) \sim \mathcal{D}_b}[\|\nabla_{\boldsymbol{x}} \ell(f_g(\boldsymbol{x}), y)\|]$$

$$+ \frac{1}{\epsilon} \cdot \mathbb{E}_{(\boldsymbol{x},y) \sim \mathcal{D}_b}[\ell_{trans}(f_g, f_p, \boldsymbol{x}, y)] + \lambda \cdot \epsilon \Big)$$

*Proof.* According to Lemma 2, we know:

$$\ell_{trans}(f_g, f_p, \boldsymbol{x}, y) \geq \epsilon \cdot \|\nabla_{\boldsymbol{x}} \ell(f_g(\boldsymbol{x}), y)\| \cdot (1 - \frac{1}{\gamma} \cdot \cos\theta) - \lambda \cdot \epsilon^2$$

where $\cos\theta = \frac{\nabla_{\boldsymbol{x}} \ell(f_g(\boldsymbol{x}),y) \nabla_{\boldsymbol{x}} \ell(f_p(\boldsymbol{x}),y)}{\|\nabla_{\boldsymbol{x}} \ell(f_g(\boldsymbol{x}),y)\| \|\nabla_{\boldsymbol{x}} \ell(f_p(\boldsymbol{x}),y)\|}$. Then, we derive an upper bound of $\|\nabla_{\boldsymbol{x}} \ell(f_g(\boldsymbol{x}), y)\| \cdot \sin\theta$ from $\ell_{trans}(f_g, f_p, \boldsymbol{x}, y)$. It is easy to see that $\theta \in [0, \frac{\pi}{4}]$. Therefore, we have:

$$\ell_{trans}(f_g, f_p, \boldsymbol{x}, y)$$

$$\geq \epsilon \cdot \|\nabla_{\boldsymbol{x}}\ell(f_g(\boldsymbol{x}), y)\| \cdot (1 - \frac{1}{\gamma} \cdot \cos\theta) - \lambda \cdot \epsilon^2$$

$$= \frac{\epsilon}{\gamma} \cdot \|\nabla_{\boldsymbol{x}}\ell(f_g(\boldsymbol{x}), y)\| \cdot (1 - \cos\theta + \gamma - 1) - \lambda \cdot \epsilon^2$$

$$= \frac{\epsilon}{\gamma} \cdot \|\nabla_{\boldsymbol{x}}\ell(f_g(\boldsymbol{x}), y)\| \cdot (1 - \cos\theta) + \frac{\epsilon \cdot (\gamma - 1)}{\gamma} \cdot \|\nabla_{\boldsymbol{x}}\ell(f_g(\boldsymbol{x}), y)\| - \lambda \cdot \epsilon^2$$

$$= \frac{\epsilon}{\gamma} \cdot \|\nabla_{\boldsymbol{x}}\ell(f_g(\boldsymbol{x}), y)\| \cdot (2 \cdot \sin^2\frac{\theta}{2}) + \frac{\epsilon \cdot (\gamma - 1)}{\gamma} \cdot \|\nabla_{\boldsymbol{x}}\ell(f_g(\boldsymbol{x}), y)\| - \lambda \cdot \epsilon^2$$

$$= \frac{\epsilon}{\gamma} \cdot \|\nabla_{\boldsymbol{x}}\ell(f_g(\boldsymbol{x}), y)\| \cdot (\sin\theta + 2 \cdot \sin^2\frac{\theta}{2} - \sin\theta)$$

$$+ \frac{\epsilon \cdot (\gamma - 1)}{\gamma} \cdot \|\nabla_{\boldsymbol{x}}\ell(f_g(\boldsymbol{x}), y)\| - \lambda \cdot \epsilon^2$$

$$= \frac{\epsilon}{\gamma} \cdot \|\nabla_{\boldsymbol{x}}\ell(f_g(\boldsymbol{x}), y)\| \cdot \sin\theta + \frac{\epsilon}{\gamma} \cdot \|\nabla_{\boldsymbol{x}}\ell(f_g(\boldsymbol{x}), y)\| \cdot (2 \cdot \sin^2\frac{\theta}{2} - \sin\theta)$$

$$+ \frac{\epsilon \cdot (\gamma - 1)}{\gamma} \cdot \|\nabla_{\boldsymbol{x}}\ell(f_g(\boldsymbol{x}), y)\| - \lambda \cdot \epsilon^2$$

$$\geq \frac{\epsilon}{\gamma} \cdot \|\nabla_{\boldsymbol{x}}\ell(f_g(\boldsymbol{x}), y)\| \cdot \sin\theta + \frac{\epsilon}{\gamma} \cdot \|\nabla_{\boldsymbol{x}}\ell(f_g(\boldsymbol{x}), y)\| \cdot (1 - \sqrt{2})$$

$$+ \frac{\epsilon \cdot (\gamma - 1)}{\gamma} \cdot \|\nabla_{\boldsymbol{x}}\ell(f_g(\boldsymbol{x}), y)\| - \lambda \cdot \epsilon^2$$

$$= \frac{\epsilon}{\gamma} \cdot \|\nabla_{\boldsymbol{x}}\ell(f_g(\boldsymbol{x}), y)\| \cdot \sin\theta + \frac{\epsilon \cdot (\gamma - \sqrt{2})}{\gamma} \cdot \|\nabla_{\boldsymbol{x}}\ell(f_g(\boldsymbol{x}), y)\| - \lambda \cdot \epsilon^2$$

Moving $\|\nabla_{\boldsymbol{x}}\ell(f_g(\boldsymbol{x}), y)\| \cdot \sin\theta$ to the left hand side (LHS):

$$\|\nabla_{\boldsymbol{x}}\ell(f_g(\boldsymbol{x}), y)\| \cdot \sin\theta$$
$$\leq \frac{\gamma}{\epsilon} \cdot \ell_{trans}(f_g, f_p, \boldsymbol{x}, y) + (\sqrt{2} - \gamma) \cdot \|\nabla_{\boldsymbol{x}}\ell(f_g(\boldsymbol{x}), y)\| + \gamma \cdot \lambda \cdot \epsilon \tag{12}$$

According to Lemma 1, we know:

$$\mathcal{L}(f_p, \mathcal{D}_{bc}) - \mathcal{L}(f_p, \mathcal{D}_b) = \mathbb{E}_{(\boldsymbol{x}, y) \sim \mathcal{D}_b, \boldsymbol{\Delta} \sim \mathcal{D}_{\boldsymbol{\Delta}|\boldsymbol{x}}}[\int_{\alpha=0}^1 \langle \nabla_{\boldsymbol{x}_s}\ell(f_p(\boldsymbol{x} + \alpha \cdot \boldsymbol{\Delta}), y), \mathbf{1}\rangle \mathrm{d}\alpha]$$
$$< \mathbb{E}_{(\boldsymbol{x}, y) \sim \mathcal{D}_b}[\frac{\sqrt{d_s}}{\gamma} \cdot \|\nabla_{\boldsymbol{x}}\ell(f_g(\boldsymbol{x}), y)\| \cdot (\sin\theta_g + \sin\theta)] \tag{13}$$

Combining Eq. (13) and Eq. (12), and taking expectation of $\boldsymbol{x}$, $y$ over $\mathcal{D}_b$:

$$\mathbb{E}_{\boldsymbol{x} \sim \mathcal{D}_b}[\frac{\sqrt{d_s}}{\gamma} \cdot \|\nabla_{\boldsymbol{x}}\ell(f_g(\boldsymbol{x}), y)\| \cdot (\sin\theta_g + \sin\theta)]$$
$$\leq \frac{\sqrt{d_s}}{\gamma} \cdot \left( \mathbb{E}_{(\boldsymbol{x}, y) \sim \mathcal{D}_b}[\|\nabla_{\boldsymbol{x}}\ell(f_g(\boldsymbol{x}), y)\| \cdot \sin\theta_g] + \mathbb{E}_{(\boldsymbol{x}, y) \sim \mathcal{D}_b}[\frac{\gamma}{\epsilon} \cdot \ell_{trans}(f_g, f_p, \boldsymbol{x}, y)] \right.$$
$$\left. + \mathbb{E}_{(\boldsymbol{x}, y) \sim \mathcal{D}_b}[(\sqrt{2} - \gamma) \cdot \|\nabla_{\boldsymbol{x}}\ell(f_g(\boldsymbol{x}), y)\|] + \gamma \cdot \lambda \cdot \epsilon \right) \tag{14}$$
$$\leq \sqrt{d_s} \cdot \left( (\frac{\sin\theta_g + \sqrt{2}}{\gamma_{\min}} - 1) \cdot \mathbb{E}_{(\boldsymbol{x}, y) \sim \mathcal{D}_b}[\|\nabla_{\boldsymbol{x}}\ell(f_g(\boldsymbol{x}), y)\|] \right.$$
$$\left. + \frac{1}{\epsilon} \cdot \mathbb{E}_{(\boldsymbol{x}, y) \sim \mathcal{D}_b}[\ell_{trans}(f_g, f_p, \boldsymbol{x}, y)] + \lambda \cdot \epsilon \right)$$

Plugging Eq. (14) back to Eq. (13) completes the proof.

$\square$

## C    More Experimental Setting

### C.1    Data Partition

We distribute the MNIST and Coil20 dataset across 50 clients. Each client has data from 5 different classes. The local dataset on each client is further partitioned to train/validation/test set with a ratio of 72:8:20, following prior work (Li et al., 2021). The test set here is biased. We alternate the spurious features in biased test sets by recoloring the data to create a bias-conflicting test set. For the CelebA dataset, we consider two partitions. In the first partition (CelebA_R), each client represents 20 celebrities. One celebrity only appears on one client. The blond male images in the biased test sets are copied to bias-conflicting test sets. We use all clients for training the global model. In the personalization step, we select the clients who have more than 5 blond female training samples and more than 5 blond male test samples. We select these clients because they provide enough samples to create spurious correlations and bias-conflicting test sets. Although the first partition on CelebA is real, the number (161) of blond male images is small. To make the result clearer, we create another synthetic CelebA partition. In the second partition (CelebA_S), there are 650 blond male images, which achieve a similar bias-conflicting test set size as prior works Sagawa et al. (2020); Liu et al. (2021). The 650 images are distributed to 3 clients with 2350 other images. The rest of the images are distributed in the same way as the first partition. Tables 3, 4, and 5 provide more details about the 3 clients.

Table 3: Number of Train and Validation Samples in CelebA_S

| Client ID | Non-blond Female | Non-Blond Male | Blond Female | Blond Male |
|---|---|---|---|---|
| 0 | 55 | 31 | 12 | 2 |
| 1 | 30 | 68 | 0 | 2 |
| 2 | 59 | 28 | 11 | 2 |

Table 4: Number of Biased Test Samples in CelebA_S

| Client ID | Non-blond Female | Non-Blond Male | Blond Female | Blond Male |
|---|---|---|---|---|
| 0 | 115 | 60 | 45 | 2 |
| 1 | 60 | 75 | 79 | 0 |
| 2 | 86 | 111 | 14 | 0 |

Table 5: Number of Biased-Conflicting Test Samples in CelebA_S

| Client ID | Non-blond Female | Non-Blond Male | Blond Female | Blond Male |
|---|---|---|---|---|
| 0 | 0 | 0 | 0 | 200 |
| 1 | 0 | 0 | 0 | 203 |
| 2 | 0 | 0 | 0 | 204 |

### C.2    Hyper-parameters

We use Adam optimizer (Kingma & Ba, 2015) throughout our experiments with learning rate 1e-4. Although stochastic gradient descent (SGD) optimizer is more common in vision-related tasks, we find that the Adam optimizer always leads to lower accuracy disparity. We train the global model for 500 rounds. 5 clients are selected per round, and each performs 5 epochs of local updates. We tune the coefficients of the adversarial transferability and $L_2$ regularization terms from $\{0.01, 0.1, 1.0, 10.0\}$ and select the largest value that does not decrease the validation accuracy during penalization. We start the attack budget at $0.031$ and gradually decrease it such that $30\% - 50\%$ of the attack succeeds. A large budget will make the attack too strong and push the adversarial examples far across the decision boundary, making the regularization method less effective. We configure $\epsilon$ to 0.031/0.01/0.015 for MNIST/CelebA/Coil20, respectively. We fine-tune the global model for 5 epochs on MNIST and

10 epochs on CelebA/Coil20, which are sufficient for the personalized models to converge. The clients with the most data samples fine-tune the penalized models for a total of 80/40/30 batches on MNIST/CelebA/Coil20 datasets. Note that we may not select the personalized model with the most fine-tuning batches for reporting. In the just train twice (JTT) method, we up-sample the residuel by a factor of 50. In Ditto, we tune its $\lambda$ from $\{0.1, 1.0\}$.

### C.3 Neural Network Architecture

We use CNN 28x28 for MNIST dataset and CNN 64x64 for CelebA and Coil20 datasets.

Table 6: Neural Network Architecture

| CNN 28x28 | CNN 64x64 |
|---|---|
| Input: $\mathbb{R}^{3 \cdot 28 \cdot 28}$ | Input: $\mathbb{R}^{3 \cdot 64 \cdot 64}$ |
| 4·4 conv, 64 BN LReLU, stride 2 | 4·4 conv, 64 BN LReLU, stride 2 |
| 4·4 conv, 128 BN LReLU, stride 2 | 4·4 conv, 64 BN LReLU, stride 2 |
| FC 4096 ReLU | FC 4096 ReLU |
| FC 10 | FC 10 |

## D More Experimental Results

### D.1 First-order Approximation of Adversarial Transferability Loss

To explore the impact of the $\lambda$ term in Lemma 2 and Theorem 3, we measure the relative difference between $\ell_{trans}(f_g, f_p, \boldsymbol{x}, y)$ and $\epsilon \cdot \|\nabla_{\boldsymbol{x}} \ell(f_g(\boldsymbol{x}), y)\| \cdot (1 - \frac{1}{\gamma} \cdot \cos\theta)$. In other words, we measured the accuracy of a first-order approximation of $\ell_{trans}(f_g, f_p, \boldsymbol{x}, y)$. If the approximation is accuracy is high, it implies that the impact of $\lambda \cdot \epsilon^2$ is low. Specifically, we compute an approximation error:

$$\mathbb{E}_{(\boldsymbol{x}, y) \sim \mathcal{D}_b} \left[ \left| \frac{\ell_{trans}(f_g, f_p, \boldsymbol{x}, y) - \epsilon \cdot \|\nabla_{\boldsymbol{x}} \ell(f_g(\boldsymbol{x}), y)\| \cdot (1 - \frac{1}{\gamma} \cdot \cos\theta)}{\ell_{trans}(f_g, f_p, \boldsymbol{x}, y)} \right| \right].$$

On MNIST, CelebA and Coil20 datasets, we find that the approximation error is 0.019, 0.058, 0.084, respectively. These results suggests that using $\epsilon \cdot \|\nabla_{\boldsymbol{x}} \ell(f_g(\boldsymbol{x}), y)\| \cdot (1 - \frac{1}{\gamma} \cdot \cos\theta)$ to approximate $\ell_{trans}(f_g, f_p, \boldsymbol{x}, y)$ results in a decent accuracy and the impact of $\lambda \cdot \epsilon^2$ is low. The possible reason is that the attack budget $\epsilon$ is usually small (e.g., 0.031), such that the gradient of a function changes little in a small neighborhood defined by $\epsilon$.

### D.2 Effectiveness of $L_2$ Regularization Term

Figure 7 shows the distribution of $\gamma$ before and after applying the $L_2$ regularization term on the CelebA dataset. Here, the $\gamma$ is computed once per data sample. We keep the global model fixed and fine-tune the personalized model for 1 epoch.

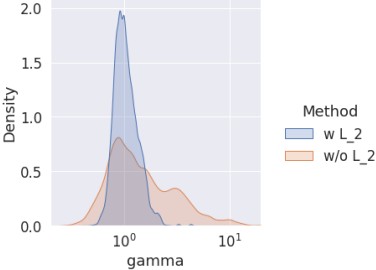

Figure 7: With the $L_2$ regularization term, the distribution of $\gamma$ is centered around 1, with a small variance. The minimum of $\gamma$ is closer to 1.

### D.3 DIVERSITY OF BIAS-CONFLICTING SAMPLES IMPACTS DEBIASING

To explore the impact of the diversity of bias-conflicting samples on debiasing, we vary the diversity of bias-conflicting samples and adjust the up-weighting factors accordingly. Specifically, we sample a factor of 0.02, 0.025, 0.033, 0.05, and 0.1 biased data samples from the MNIST dataset and re-color them to become bias-conflicting. The factor in the sampling step is called sampling factor. Then, we up-weight the bias-conflicting samples by a factor of 50, 40, 30, 20, 10, respectively, keeping the total number of bias-conflicting samples consistent. Here, the bias-conflicting samples have less diversity if generated by a small number of biased data samples with a large up-weighting factor. Experimental results in Figure 8 show that, as the diversity reduces, the accuracy disparity of personalized model on the biased dataset and bias-conflicting dataset increases, supporting our analysis. Therefore, our method is applicable in the scarcity of bias-conflicting samples while the up-weighting method fails.

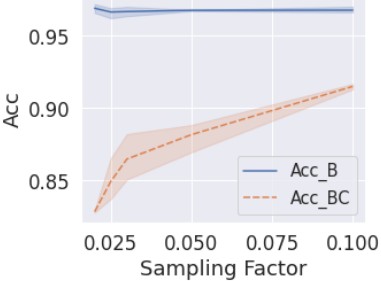

Figure 8: The up-weighting method is less effective, resulting in large accuracy disparity of personalized model on biased dataset and bias-conflicting dataset, if the bias-conflicting samples is generated by a small number of biased data samples using a small sampling factor and a large up-weighting factor. The up-weighting factor is set to be the reciprocal of the sampling factor.

