# OpenReview forum: "Robust and Personalized Federated Learning with Spurious Features: an Adversarial Approach"
_ICLR.cc/2022/Conference — ICLR 2022 Submitted_

### Official Review · Reviewer_CHgD · 2021-10-31

**Correctness:** 3
**Technical Novelty And Significance:** 2
**Empirical Novelty And Significance:** Not applicable
**Recommendation:** 3
**Confidence:** 4

**Main Review:**

Strength:
- The idea of using adversarial examples and minimizing entropy loss of global and local models' outputs is normal. The intuitions are straightforward to understand.
- Authors have spent a decent amount of effort explaining the relationship between the spurious features and adversarial transferability. It is helpful for audiences.
- The results do show certain improvements over the other baselines.

Weakness:
- The presentation has space for improvement, please explicitly explain all the critical terms (accuracy disparity, bias-conflicting, etc) at the first occurrence in this paper.
- Many subjective descriptions exist in the paper, e.g., if you claim the distribution shift of spurious features is a major effect of accuracy disparity, it would be really important that you give theoretical proof and/or empirical support to verify your claim.
- It seems like the spurious features are handcrafted, and we don't have a clear solution of how to automatically choose the spurious features in real applications.
- The adversarial examples are generated using a global model, however, the way of generating adversarial examples in FL worth a lot of analysis and description of the details. There exist some papers that discuss the best way of improving adversarial robustness with adversarial training. Here, the same strategy should be applied to compare.
- The experiments are a bit disappointing. Without a comprehensive comparison, a replication of the results is almost impossible. So many factors in FL can dramatically change the results. The authors didn't provide a fair and reproducible setting for the results. The experiment is incomplete and the results are not convincing.



**Summary Of The Paper:**

The authors have proposed a new FL training strategy to reduce the performance discrepancy between the central model and the client models. The PGD generated adversarial examples are fed into both central model and client models and their outputs are used to minimize the entropy loss. The computation is further simplified using Taylor expansion.

**Summary Of The Review:**

The idea is good and novel, however, the presentation is disappointing and the experiments are weak. I recommend rejection.

---

> ### Author Response · Authors · 2021-11-17
> **Responses to Reviewer CHgD**
>
> We thank the reviewer for acknowledging the novelty of our work and the improvements in our results. Below are our responses to the comments:
>
> &nbsp;
>
> **Comment**:
>
> The presentation has space for improvement, please explicitly explain all the critical terms (accuracy disparity, bias-conflicting, etc) at the first occurrence in this paper.
>
> **Response**:
>
> We have added the definitions accordingly in the revised paper.
>
> &nbsp;
>
>
> **Comment**:
>
> Many subjective descriptions exist in the paper, e.g., if you claim the distribution shift of spurious features is a major effect of accuracy disparity, it would be really important that you give theoretical proof and/or empirical support to verify your claim.
>
> **Response**:
>
> The reviewer may have missed our experimental setup and empirical results in Section 4. We presented controlled variable experiments, showing that the distribution shift of spurious features causes the accuracy disparity. The controlled variable here is the spurious features. The accuracy disparity is defined as the accuracy difference between two datasets, which only differs on the spurious feature. This setup excludes other factors affecting the accuracy disparity and has been employed in various prior works [1, 2].
>
> Additionally, we have added theoretical justification, showing the distribution shift causes the accuracy disparity (Lemma 1) to the revised paper.
>
> &nbsp;
>
> **Comment**:
>
> It seems like the spurious features are handcrafted,
>
> **Response**:
>
> We want to remind the reviewer that the spurious feature (hair color) of the CelebA dataset is natural, not handcrafted.
>
> &nbsp;
>
> **Comment**:
>
> The adversarial examples are generated using a global model, however, the way of generating adversarial examples in FL worth a lot of analysis and description of the details. There exist some papers that discuss the best way of improving adversarial robustness with adversarial training. Here, the same strategy should be applied to compare.
>
> **Response**:
>
> The reviewer may have missed many discussions in our paper.
>
> First, we discussed two reasons for generating adversarial examples using a global model in the methods section, which are re-stated below:
>
> (1)	The global model is fixed, and the personalized model is trainable.
>
> (2)	The global model is unbiased, a desirable property that we want to achieve with the personalized model.
>
> Therefore, with reason (1), generating adversarial examples using the global model and transferring the examples to the personalized model is a natural choice. In contrast, the opposite direction does not work because the global model is not trainable. With reason (2), the adversarial examples generated by the global model encode the perturbations on the non-spurious features, which helps the personalized model choose the non-spurious features.
>
>
> Second, we discussed the way of generating adversarial examples. We focus on first-order methods because they are computationally efficient (Section 6), important for the on-device personalization step.
>
> Third, we included the description of the details of the PGD attack in the method section and reported the hyper-parameters in the experiment section.
>
> Finally, the PGD attack we employed is the strongest first-order attack in adversarial training, and training against PGD attack provides a natural and broad security guarantee [3].
>
> &nbsp;
>
>
> **Comment**:
>
> A replication of the results is almost impossible
>
> **Response**:
>
> It is disappointing to see the reviewer missed our efforts towards reproducibility in the experiment section, including reporting:
> 1. The data partition: the number of users, the number of classes per user, the open-source library that includes the data partition scripts, and the split ratio for train/validation/test sets.
> 2. The hyper-parameters: optimizer, learning rate, co-efficiencies for regularization terms, attack budgets for adversarial attacks, training rounds, number of clients per round, local epochs per round, fine-tuning epochs.
> 3. The neural network architecture.
>
> We will be glad to report any additional setup if the reviewer believes they are helpful.
>
> &nbsp;
>
> ### Reference
>
> [1] Liu, Evan Z., et al. "Just train twice: Improving group robustness without training group information." International Conference on Machine Learning. PMLR, 2021.
>
> [2] Sagawa, Shiori, et al. "Distributionally robust neural networks for group shifts: On the importance of regularization for worst-case generalization." arXiv preprint arXiv:1911.08731 (2019).
>
> [3] Madry, Aleksander, et al. "Towards Deep Learning Models Resistant to Adversarial Attacks." International Conference on Learning Representations. 2018.

---

### Official Review · Reviewer_eYjN · 2021-11-02

**Correctness:** 4
**Technical Novelty And Significance:** 3
**Empirical Novelty And Significance:** 3
**Recommendation:** 6
**Confidence:** 3

**Main Review:**

Strength
- The work exposes a possible generalisation issues in personalised federated learning and proposes a novel approach to tackle it
- The idea is well motivated, paper is generally well written and experiments are provided to substantiate the claims

Weakness
- Use of some non-standard hyperparams like 0.031 eps budget for MNIST and the batches of 96, 40, 30. Similarly the 5 epochs of local training seem larger than the conventional 1 or 2. Could the authors provide an explanation?
- Doesn’t include exploration for other modalities like text or large scale setups like FEMNIST

Few open questions
- Any thoughts on how the method could compare to say the adversarial training objective in combination with personalisation?


**Summary Of The Paper:**

This work explores the possibility of personalisation methods entangling spurious features that can undermine their generalization in case of federated learning. It proposes to use a combination of a consistency term for adversarial transferability and an L2 regularisation term to help reduce this disparity.  The approach is evaluated on artificial settings with spurious features.

**Summary Of The Review:**

I think this work tackles an interesting hypothesis that can limit generalization in case of personalised FL. The proposed solution is principled and well motivated with appropriate ablation study. The only drawback would be lack of experimentation on large scale problems which would certainly make this a valuable piece of work.

---

> ### Author Response · Authors · 2021-11-17
> **Responses to Reviewer eYjN**
>
> We thank the reviewer for spending time reviewing our paper, making helpful comments, and acknowledging the novelty, motivation, writing, and experimental results of our paper! Below are our responses to the concerns that the reviewer raises.
>
> &nbsp;
>
> **Comment**:
>
> Use of some non-standard hyperparams like 0.031 eps budget for MNIST and the batches of 96, 40, 30. Similarly the 5 epochs of local training seem larger than the conventional 1 or 2.
>
> **Response**:
>
> Among various prior works and benchmarks [1,2], the attack budget of 0.031 (a.k.a 8/255) is one of the most common choices. Therefore, we chose 0.031 as a starting point.
>
> We thank the reviewer for pointing out the issue regarding the batches. The paper is revised accordingly: We fine-tune the global model for 5 epochs on MNIST and 10 epochs on CelebA/Coil20, which are sufficient for the personalized models to converge. The clients with the most data samples fine-tune the penalized models for a total of 80/40/30 batches on MNIST/CelebA/Coil20 datasets
>
> We checked out the FedAVG paper [3] and found that setting the local training epoch to 5 grants faster convergence of the global model.
>
> &nbsp;
>
> **Comment**:
>
> Doesn’t include exploration for other modalities like text or large scale setups like FEMNIST
>
> **Response**:
>
> Our experiments on the CelebA dataset are larger-scale than prior works on the FEMNIST dataset in terms of the client setup. We distribute the data across 508 clients according to the prior work [4]. In comparison, the prior setup [4] for the FEMNIST dataset only has 205 clients.
>
> Since the adversarial attack for text data is quite different from the attack for visual data and is often studied separately [5], we plan to study text data in the future.
>
> &nbsp;
>
> **Comment**:
>
> Any thoughts on how the method could compare to say the adversarial training objective in combination with personalisation?
>
> **Response**:
>
> We did have experiments using adversarial training. However, we did not find any improvement. The reason is that adversarial training does not improve the similarity between the global and personalized models. In contrast, using our method to enforce adversarial transferability enforces the personalized model to use the same set of non-spurious features as the global model uses.
>
> &nbsp;
>
> ### Reference
> [1] Madry, Aleksander, et al. "Towards Deep Learning Models Resistant to Adversarial Attacks." International Conference on Learning Representations. 2018.
>
> [2] Croce, Francesco, et al. "Robustbench: a standardized adversarial robustness benchmark." arXiv preprint arXiv:2010.09670 (2020).
>
> [3] McMahan, Brendan, et al. "Communication-Efficient Learning of Deep Networks from Decentralized Data." Artificial Intelligence and Statistics. PMLR, 2017.
>
> [4] Li, Tian, et al. "Ditto: Fair and robust federated learning through personalization." International Conference on Machine Learning. PMLR, 2021.
>
> [5] Zhang, Wei Emma, et al. "Adversarial attacks on deep-learning models in natural language processing: A survey." ACM Transactions on Intelligent Systems and Technology (TIST) 11.3 (2020): 1-41.

---

### Official Review · Reviewer_jRKj · 2021-11-03

**Correctness:** 2
**Technical Novelty And Significance:** 2
**Empirical Novelty And Significance:** 2
**Recommendation:** 3
**Confidence:** 4

**Main Review:**

**Strengths**

- `The paper identifies a relevant problem.` The paper empirically shows that prior work on personalized federated learning induces high accuracy disparity, which is a relevant problem to the fairness community.
- `The method is somewhat novel and technically sound.` The paper demonstrates a link between adversarial transferability and accuracy disparity of personalized models. Thus, the paper employs adversarial transferability and weight regularization (which has already been proposed by prior work, as the paper acknowledges) to learn personalized federated models that achieve high accuracy and lower accuracy disparity than prior work.

**Weaknesses**

- `Certain presentational aspects could be improved.` For example, "bias-conflicting examples" are mentioned in the introduction but only defined in section 4 (leaving the reader to guess what these "bias-conflicting examples" are). Furthermore, the experimental setup for figures 4 and 5 is not provided (e.g., which personalization method was used?). Moreover, in section 6.1 two sentences are repeated (“there are 650 blond…” and “there are 650 blond...”). Finally, there are a few typos (Section 4.1 “models on bias**ed** and bias-conflicting”, section 6.1 “through**out** our experiments”, section 6.1 “residu**a**l”).
- `The method lacks key motivation.` Adversarial transferability is introduced to reduce the accuracy disparity of the personalized models by forcing them to be vulnerable to the same adversarial examples as the global model. However, the adversarial transferability between the global and local models is just a proxy for the similarity of the two models, similar to, e.g., KL divergence.
- `Baselines are misrepresented.` In figure 3, the centralized model is trained on a dataset where the spurious correlations are fixed, whereas the federated model is trained on multiple client datasets, each with different spurious correlations. Accordingly, it is unsurprising that the centralized model has a significantly lower accuracy on the bias-conflicting dataset, as it was not exposed to the same data distribution shifts during training as the federated model. I would expect the centralized model to perform on par with the federated model when trained on the same data, invalidating the paper’s claim.
- `Results are misrepresented.` In section 5, the paper states that low adversarial transferability indicates high accuracy disparity for the personalized models. However, the paper merely shows a correlation between the two metrics (as they both estimate the similarity between the global and personalized models). I would be very surprised if one could not train global and personalized models that achieve “high transferability and high disparity” or “low transferability but high disparity” (as I believe that transferability and accuracy disparity are only weakly correlated). Unfortunately, the paper does not provide empirical evidence to substantiate these claims. (In fact, it provides evidence for the opposite as “accuracy disparity still increases, even if the adversarial transferability remains high”).
- `The paper makes unsubstantiated claims.` In section 5, the paper claims that “Both methods are relatively light-weight from a computational perspective” but does not provide any further evidence for or analysis of this statement.
- `Results are unclear.` Figure 7 compares the losses on the biased and bias-conflicting datasets. However, the loss is only an approximation of the accuracy, which is the quantity that we are ultimately interested in. Therefore, it would be more meaningful to compare the accuracies on the different datasets.
- `Results are insignificant.` Comparing the results for the global model and the proposed method in table 1, the method achieves roughly the same accuracy on the bias-conflicting dataset and only a minor increase in accuracy on the biased dataset (except for the MNIST dataset). Moreover, the prior personalization approaches achieve significantly higher accuracies on the biased datasets. Thus, the method just provides a “little less” personalization, yielding personalized models that are more similar to the global model (with corresponding performance).

**Summary Of The Paper:**

The paper considers the problem that personalization methods in federated learning may cause the personalized models to overfit on spurious features, thereby increasing the accuracy disparity compared to the global model. To mitigate this accuracy disparity, the paper investigates adversarial transferability, which is shown to correlate with disparity. Thus, the paper proposes a federated personalization approach based on adversarial transferability and catastrophic forgetting that reduces accuracy disparity to the level of the global model while maintaining the higher accuracy of prior personalization methods. The paper evaluates the approach on three real-world datasets.

**Summary Of The Review:**

The paper identifies an interesting shortcoming of prior personalization approaches. However, given the various weaknesses outlined above (e.g., insignificant results, misrepresentation of results) and the limited novelty, I do not believe that the paper meets the bar for publication in its current form.

---

> ### Author Response · Authors · 2021-11-17
> **Responses to Reviewer jRKj (2/2)**
>
> **Comment**:
>
> Results are insignificant. The prior personalization approaches achieve significantly higher accuracies on the biased datasets
>
> **Response**:
>
> The reviewer may have forgotten our study's backgrounds (i.e., fairness and robustness, as the reviewer acknowledges), which are stated in Section 1, and misinterpreted our experimental results. The unique advantage of our method is to improve the accuracy by an average of 3.43% (Table 1) on biased datasets without sacrificing the accuracy disparity or causing issues in fairness and robustness.
>
> The significantly higher accuracy increase of prior personalization approaches on biased dataset is the evidence for producing biased models, which are expected to perform better than unbiased models on biased datasets. Additional evidence is that the prior personalization approaches, when paired with debiasing methods (i.e., up-weighting, JTT), have similar performance on biased datasets to our methods (Table 1). These results suggest that our method is comparable with prior approaches on biased datasets if those approaches do not entangle spurious features.
>
> The accuracy benefits of biased models on biased datasets may mislead users (e.g., the reviewer) to choose biased models, which increases the accuracy disparity by up to 15.12% (Figure 4, Table 1), resulting in issues in both fairness and robustness (Section 1).
>
> &nbsp;
>
> ### Reference
>
> [1] Tramèr, Florian, et al. "The space of transferable adversarial examples." arXiv preprint arXiv:1704.03453 (2017).
>
> [2] Liang, Kaizhao, et al. "Uncovering the connections between adversarial transferability and knowledge transferability." International Conference on Machine Learning. PMLR, 2021.

---

> ### Author Response · Authors · 2021-11-17
> **Responses to Reviewer jRKj (1/2)**
>
> We thank the reviewer for acknowledging the relevance of the problem and the novelty and soundness of our method. Below are our responses to the comments:
>
> &nbsp;
>
> **Comment**:
>
> The paper employs adversarial transferability and weight regularization (which has already been proposed by prior work.
>
> **Response**:
>
> The reviewer may have misunderstood the prior works and misinterpreted our methods, which are proposed in this paper. Our work differs from prior works in the following three aspects:
> 1. We show a new and relevant problem (as the reviewer acknowledges) that could be solved by enforcing adversarial transferability. In contrast, prior works either studied adversarial transferability alone [1] or studied a different problem (e.g., transfer learning [2]).
> 2. We make the adversarial transferability a training objective, which any prior work has not explored, and we show its benefit (lower accuracy disparity, as the reviewer acknowledges).
> 3. We complement our method by a weight regularization term (as the reviewer notices) after finding that enforcing the adversarial transferability alone is insufficient. This issue is also unexplored.
>
> &nbsp;
>
> **Comment**:
>
> Certain presentational aspects could be improved. For example, "bias-conflicting examples" are mentioned in the introduction but only defined in section 4 (leaving the reader to guess what these "bias-conflicting examples" are). Furthermore, the experimental setup for figures 4 and 5 is not provided (e.g., which personalization method was used?).
>
> **Response**:
>
> We have moved the definitions to the introduction in the revised paper.
>
> The personalization method is fine-tuning, which is stated in the preliminary section. In the revised paper, we have explicitly mentioned the fine-tuning method in Section 4, where Figures 4 and 5 are located, in case the readers miss any preliminary.
>
> The typos are fixed.
>
> &nbsp;
>
> **Comment**:
>
> The method lacks key motivation. … the adversarial transferability between the global and local models is just a proxy for the similarity of the two models.
>
> **Response**:
>
> We provided intuitive and empirical motivation for our method in Section 4. We have further provided theoretical motivation for our method in Section 5 of the revised paper.
>
> We are glad to see that the reviewer noticed that the adversarial transferability between the global and local models is a proxy for the similarity of the two models. We show that the proxy is theoretically sound (Section 5 of the revised paper) and helpful in measuring the accuracy disparity, which is the problem that this paper studies.
>
>
> &nbsp;
>
> **Comment**:
>
> Baselines are misrepresented. … I would expect the centralized model to perform on par with the federated model when trained on the same data, invalidating the paper’s claim.
>
> **Response**:
>
> The reviewer may have missed our experiments on the CelebA dataset, whose setup is in Section 4. We use the same dataset to train the federated model and the centralized model. The federated model shows lower accuracy disparity.
>
> &nbsp;
>
> **Comment**:
>
> Results are misrepresented
>
> **Response**:
>
> We have a hard time understanding this self-contradictory comment. The reviewer criticized our paper for providing no empirical evidence for claiming the connection between the two metrics--the adversarial transferability and the accuracy disparity--after saying, “the paper merely shows a correlation between the two metrics.”
>
> First, the reviewer may have missed the empirical evidence in Sections 4.3 and 7.1 that supports the connection between the adversarial transferability and the accuracy disparity.
>
> In addition, the reviewer may have misinterpreted our empirical result and missed the discussion of the empirical result in Section 7.1. The case where "the accuracy disparity still increases, even if the adversarial transferability remains high" is an issue that we have already identified by only enforcing adversarial transferability. Such an issue further motivates our weight regularization method.
>
> &nbsp;
>
> **Comment**:
>
> The paper makes unsubstantiated claims
>
> **Response**:
>
> The reviewer may have missed our discussion in Section 5.1, where we mentioned that the PGD attack uses the neural network’s first-order gradient, which is easy to compute. In the revised paper, we have included additional discussions: The adversarial examples are computed using the global model once for all. The computation only needs a few back-propagations, much less than that of training the global model.
>
> &nbsp;
>
> **Comment**:
>
> Results are unclear. … the loss is only an approximation of the accuracy, which is the quantity that we are ultimately interested in.
>
> **Response**:
>
> We have added additional results under the accuracy metric to Figure 6, showing that our method is effective.

---

### Author Response · Authors · 2021-11-17
**Revision Summary**

We thank all the reviewers for their comments. The paper is revised, and the revision summary is listed below:

- We provide additional theoretical analysis in Section 5 and the proofs in Appendix B, showing that the disparity and the transferability are connected.
- The section of the weight regularization term is revised and connected to the theoretical analysis.
- Some definitions in the introduction and setup in the experiments are clarified.

---

### Decision · Program_Chairs · 2022-01-20

**Decision:**

Reject

**Comment:**

The paper talks about a novel setting in Federated Learning and argues that personalization methods may cause the personalized models to overfit on spurious features, thereby increasing the accuracy disparity compared to the global model. To this end the authors propose a debiasing strategy using a global model and adversarial tranferability.

 There were some positive opinion about the problem being interesting .However reviewers had several concerns about the validity of assumption and hand wavy arguments used in the solutions for existence adversarial tranferability. Overall, the settings and the need for removing personalization bias needs to be validated more convincingly and rigorously, with concrete real scenarios and experiments.